# ANY-PROPERTY-CONDITIONAL MOLECULE GENERATION WITH SELF-CRITICISM USING SPANNING TREES

## ABSTRACT

Generating novel molecules is challenging, with most representations of molecules leading to generative models producing many invalid molecules. Spanning Tree-based Graph Generation (STGG) (Ahn et al., 2021) is a promising approach to ensure the generation of valid molecules, outperforming state-of-the-art generative models (Weininger, 1988; Song & Ermon, 2019) for unconditional generation. In the real world, we want to be able to generate molecules conditional on one or multiple desired properties rather than unconditionally. Thus, in this work, we extend STGG to multi-property conditional generation. Our approach, **STGG+**, incorporates a modern Transformer architecture, random masking of properties during training (enabling conditioning on *any* subset of properties and classifier-free guidance), an auxiliary property-prediction loss (allowing the model to *self-criticize* molecules and select the best ones), and other improvements. We show that **STGG+** achieves state-of-the-art performance on in-distribution and out-of-distribution conditional generation, as well as reward maximization.

## 1 INTRODUCTION

Generating novel molecules is challenging, and the choice of molecular representation significantly impacts the performance of generative models. Traditional methods have mainly focused on SMILES (Weininger, 1988) 1D strings (Segler et al., 2018; Kwon et al., 2023), and 2D graphs (Jo et al., 2022; Vignac et al., 2022; Jo et al., 2023). A significant issue with these representations is that a single error by the generative model can result in invalid molecules, especially as molecule size increases.

Recently, Krenn et al. (2020) proposed Self-referencing Embedded Strings (SELFIES) (Krenn et al., 2020), a robust 1D string representation similar to SMILES that guarantees the generation of valid molecules through a carefully designed context-free grammar. However, recent work by Gao et al. (2022) and Ghugare et al. (2023) found that while SELFIES prevent invalid molecules, it makes exploration more difficult and reduces the performance of generative models (in terms of obtaining high-reward samples, i.e., molecules with desired properties). A significant challenge for the generative models based on SELFIES is the need to pre-define the number of tokens contained in a branch (a deviation from the main path in a 1D string) and count backward the number of tokens required to reach the beginning of the ring (starting from the end). This requires extensive planning and counting, making the problem much more challenging for the model to solve.

An alternative approach, Spanning Tree-based Graph Generation (STGG) (Ahn et al., 2021), has recently emerged. Unlike SELFIES, STGG is designed explicitly for generative models and works by masking invalid tokens during sampling, preventing the generation of invalid structures (e.g., atoms without bonds between them, branch end before branch start) and ensuring proper valency. STGG uses a simple set of if/else conditions to mask out invalid tokens, which not only prevents invalid molecules but also leads to higher-quality and more diverse generated molecules (Ahn et al., 2021). Jang et al. (2023) shows that STGG generally performs equally or better than other state-of-the-art generative models (Song & Ermon, 2019; Ho et al., 2020; Song et al., 2020) for unconditional molecule generation. However, its application in multi-property conditional settings has not been explored. We address this setting along with a few additional challenges, as discussed below.

**Any-property-conditioning** In real-world applications, we want to *generate molecules conditional on one or multiple desired properties* rather than unconditionally. Furthermore, we want to condition

on *any* subset of desirable properties without retraining the model each time that we condition on a different subset of properties.

**Self-criticism**    Another critical issue is the synthesis time for molecules, which can take weeks, or months. Thus, we cannot expect chemists to synthesize all generated molecules. Ideally, we need a way to filter the molecules that we provide to chemists. Some properties can be verified through simulations, but this can be extremely slow, and not all properties can be simulated. Another option is to rely on external property predictor models, but training, validating, and managing multiple property predictors can be troublesome. *What if the generative model could predict the properties of its own generated molecules?* This is the idea we propose here: we give the model the ability to predict properties and thus *self-criticize* its own generated molecules, allowing it to automatically filter out those with undesirable properties.

**Out-of-distribution properties**    We sometimes seek to generate novel molecules with out-of-distribution (OOD) properties in order to expand the range of our molecular knowledge. These OOD properties generally involve extreme range of values. Classifier-Free Guidance (CFG) (Ho & Salimans, 2022) is a technique to improve conditioning fidelity; we found CFG useful for in-distribution properties, but problematic for some out-of-distribution conditioning values, especially for extreme values, resulting in poor generative efficiency (% of valid, unique, and novel molecules) and conditioning fidelity. Since guidance can be beneficial to some conditioning values, but not others, we propose a solution: *random guidance* with best-of-$k$ self-filtering (described further below).

In this work, we tackle any-property-conditional molecule generation with self-criticism using an improved STGG. In doing so, we make the following contributions:

1. **Any-property-conditioning**: We use an MLP on standardized continuous features and embeddings on categorical features while randomly masking some properties during training, allowing conditional generation on any number of properties (0, 1, 2, or all) and the use of Classifier-Free Guidance (CFG) for improved performance (Section 3.2).

2. **Improved Transformer architecture**: We improve on the original Transformer architecture used in STGG by using: Flash-Attention, no bias terms, RMSProp, rotary embeddings, the SwiGLU activation, and better hyperparameters (Section 3.1).

3. **Improved Spanning-Tree**: We extend STGG to 1) allow compound structures with a new token and masking conditions, 2) prevent incomplete samples through special masking when there are too many opened branches, 3) prevent ring overflow, 4) randomize the order of the graph during training for better generalization, and 5) automatically calculate valency and adapt the token vocabulary based on the dataset (Section 3.3).

4. **Auxiliary property prediction objective**: The objective improves conditioning fidelity and enables out-of-the-box self-filtering of molecules with incorrect properties. (Section 3.5)

5. **Random guidance for extreme value conditioning**: Classifier-free guidance uses guidance $w > 1$ to improve performance (Section 3.4), but this can fail when conditioning on extreme values (which are needed to generate molecules with out-of-distribution properties). We propose using random guidance with best-of-$k$ filtering as a solution (Section 3.6).

6. **Comprehensive performance evaluation**: We demonstrate excellent performance in terms of 1) distribution learning and diversity on unconditional generation (Section 4.1), 2) distribution learning and conditioning fidelity on in-distribution (Section 4.2) and out-of-distribution (Sections 4.3 and 4.5) conditional generation, and 3) diverse and high-reward samples on reward maximization (Section 4.4).

## 2    BACKGROUND

### 2.1    1D VS 2D REPRESENTATIONS

There are many ways of representing molecules in the context of molecular generation. Some of the most popular methods are autoregressive models on 1D strings (Segler et al., 2018; Ahn et al., 2021; Kwon et al., 2023) and diffusion (Song & Ermon, 2019; Ho et al., 2020; Song et al., 2020) models on 2D graphs. While both approaches have similar sample complexity, 1D strings offer a more

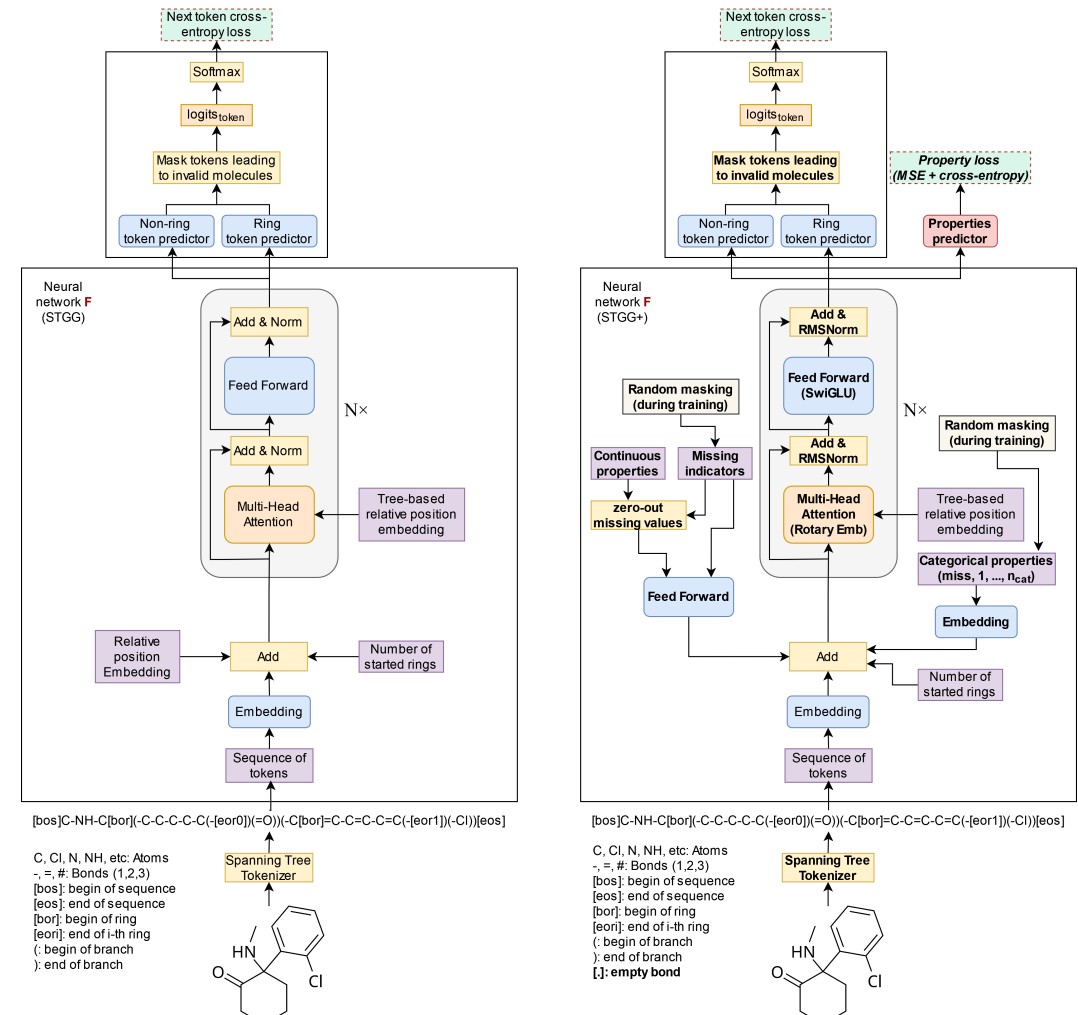

Figure 1: Left: STGG architecture, Right: Our **STGG+** architecture. The molecule is tokenized and embedded. The number of started rings and embeddings of continuous and categorical properties are added, and the output is passed to the Transformer. The Transformer output is then split to produce 1) the predicted property and 2) the token predictions (masked to prevent invalid tokens). Areas that changed or were added in **STGG+** are in **bold**. Please read the method section for more details.

compressed representation, requiring less space and fewer parameters and, thus, increased potential for scalability. We provide a detailed comparison between different representations in Appendix A.1.

Furthermore, recent results indicate that 1D strings are as competitive as 2D molecular graph methods for both unconditional molecule generation (Jang et al., 2023; Fang et al., 2023) and property prediction (Yüksel et al., 2023). Both 1D and 2D representations encapsulate the same amount of information, making the choice largely a matter of preference. We advocate for 1D string representations due to their scalability and effective utilization of Transformer models, and thus, we focus on this type of representation in our work.

## 2.2  1D STRING REPRESENTATIONS

The most popular choice of string-based representation is SMILES (Weininger, 1988), an extremely versatile method capable of representing any molecule. However, when used in generative models, generated SMILES strings often correspond to invalid molecules. A single incorrectly placed token often leads to an invalid molecule. Graph-based diffusion methods also face a similar issue. To

address this problem, recent methods like Spanning Tree-based Graph Generation (STGG) (Ahn et al., 2021) and SELFIES (Krenn et al., 2022) have been developed to prevent the generation of invalid molecules. For a detailed comparison of SMILES, SELFIES, and STGG, see Appendix A.2.

STGG has demonstrated performance on par with or better than state-of-the-art unconditional generative models (Ahn et al., 2021; Jang et al., 2023). Conversely, SELFIES has been shown to perform worse than SMILES on property-conditional molecule generation (Gao et al., 2022; Ghugare et al., 2023). Therefore, in this work, we focus on STGG for property-conditional molecule generation.

## 2.3 STGG

STGG (Ahn et al., 2021) uses a SMILES-like vocabulary with begin " (" and end ") " branch tokens, ring start "[bor]" and $i$-th ring end "[eor-$i$]" tokens. Contrary to SELFIES, STGG was made from the ground up for unconditional molecule generation. STGG leverages a Transformer (Vaswani et al., 2017) architecture to sample the next tokens conditional on the tokens of the current unfinished molecule. To predict the ring end tokens, STGG uses a similarity-based output layer distinct from the linear output layer used to predict other tokens. STGG also uses an input embedding to track the number of open rings. Invalid next tokens are prevented through masking of next tokens that would lead to impossible valencies (e.g., atoms, ring-start, and branch-start when insufficient valency remains) and structurally invalid tokens (e.g., atom after atom, bond after bond, or ring-$i$ end when fewer than $i$ ring start tokens are present).

In the next section, we will show how to improve the STGG architecture, vocabulary, and masking, adapt STGG for any-property conditional generation, and improve fidelity on conditioned properties through several techniques (classifier-free guidance, self-criticism, random classifier-free guidance for extreme conditioning).

## 3 METHOD

We tackle the problem of any-property conditional generation with self-criticism using STGG.

### 3.1 ARCHITECTURE

We enhance the architecture used in STGG, a regular Transformer (Vaswani et al., 2017) directly from PyTorch main libraries. To improve it, we leverage recent improvements in Large Language Models following GPT-3 (Radford et al., 2019), Mistral (Jiang et al., 2023), and Llama (Touvron et al., 2023). The improvements include: 1) RMSNorm (Zhang & Sennrich, 2019) replacing LayerNorm (Ba et al., 2016); 2) residual-path weight initialization (Radford et al., 2019); 3) bias-free architecture (Chowdhery et al., 2023); 4) rotary embeddings (Su et al., 2024) instead of relative positional embedding; 5) lower-memory and faster attention with Flash-Attention-2 (Dao et al., 2022; Dao, 2023); 5) SwiGLU activation function (Hendrycks & Gimpel, 2016; Shazeer, 2020); 6) changes in hyperparameters following GPT-3 (Radford et al., 2019) (i.e., AdamW (Loshchilov & Hutter, 2017; Kingma & Ba, 2014) $\beta_2 = 0.95$, cosine annealing schedule (Loshchilov & Hutter, 2016), more attention heads, no dropout). These modifications aim to enhance the model's efficiency, scalability, and overall performance. We also considered more efficient architectures, see Appendix A.5.

### 3.2 ANY-PROPERTY CONDITIONING

We preprocess continuous properties by applying a simple $z$-score standardization.

To condition the model on any subset of target properties without retraining the model every time we change the properties, we need to be able to turn off the conditioning of some properties. For continuous variables, we handle missing values through a binary indicator variable: if a property is missing, we set the property value to 0 and the missing indicator to 1. It is important to include these missing indicators because we cannot assume the plausible values for the missing features (e.g., if A is 1.0, B is missing, maybe the only possible range for B is around 3 and 4, so if we leave B at 0 without missing indicator, it will not make sense). For categorical variables, we add an extra category for missing values. During training, we mask a random subset of $t$ properties, where $t$ is chosen

uniformly between 0 and the number of properties. See Appendix A.3.2 for details. This allows us to condition the model on any subset of desired properties at test time while ignoring the rest.

In the neural network, we process the standardized continuous features (continuous properties concatenated with their binary missing indicators) in a 2-layer multilayer perceptron (MLP) with Swish activation (Hendrycks & Gimpel, 2016; Ramachandran et al., 2017). Each categorical feature is then processed individually using a linear embedding. These processed outputs are added directly to the embedding of all tokens. We also experimented with injecting these embeddings through adaptive normalization (Huang & Belongie, 2017), a method commonly used for conditioning on noise-level in diffusion models (Ho et al., 2020), but this approach massively increased the number of parameters without improving performance.

### 3.3 IMPROVEMENTS TO SPANNING-TREE

Starting from STGG as base, we implement several improvements. Firstly, we extend the vocabulary to allow for the generation of molecular compounds that are composed of multiple unconnected graphs (e.g., salt is represented as [Na+].[Cl-], where [Na+] and [Cl-] are single-atom molecules connected through a ionic bond), enabling the model to solve a broader range of problems. STGG uses a fixed vocabulary and a fixed set of maximum valencies that determines how many valence bonds each atom can form. Instead of requiring a predefined vocabulary, we automate the process of building a vocabulary based on the atoms found in the dataset and their maximum valency, again derived from the dataset. This data-centric approach allows us to represent complex structures, including non-molecular compounds containing metals.

We observe that STGG can occasionally generate incomplete samples by creating too many branches without closing them within the allowed maximum length, particularly when conditioning on extreme out-of-distribution properties. To address this, we modify the token masking process to ensure the model closes its branches when the number of open branches approaches the number of tokens left to reach the maximum length. This additional masking step prevents the rare but problematic situation of incomplete samples. Additionally, for massive molecules, it is possible for the model to rarely produce more rings than the maximum number of rings (100); we now mask the creation of rings when the maximum number is reached. With these additional masks, we generally maintain 100% validity, even when generating molecules with out-of-distribution properties).

Contrary to STGG, we do not canonicalize molecules and instead use a random ordering of the molecules (a different random ordering is sampled for each molecule during training). Doing so improves generalization (see Appendix A.8).

### 3.4 CLASSIFIER-FREE GUIDANCE

To enforce better conditioning of the properties, we use classifier-free guidance, originally designed for diffusion models (Ho & Salimans, 2022), and found beneficial for autoregressive language models as well (Sanchez et al., 2023). This technique involves directing the model more toward the conditional model's direction while pushing it away from the unconditional model's direction by an equal amount. Figure 2 illustrates this concept. The amount of guidance typically requires hyperparameter-tuning. However, for simplicity and generality, in all analyses, we arbitrarily set the guidance parameter $w$ to 1.5, where $w = 1$ means no guidance.

### 3.5 SELF-CRITICISM

To make the model more powerful, we provide the model with the ability to self-criticise its own generated molecules. The purpose is improve the quality of generated samples by using a jointly-trained property predictor to rank and filter the generated samples. It works as follows: 1) the model generates $k$ molecules for a given set of properties, 2) it evaluates the $k$ molecules molecules properties based on its own property-predictor (see the paragraph below), and 3) it returns the molecule whose properties best match the conditioned properties. This best-out-of-$k$ strategy significantly improves the quality of its generated molecules.

For the model to be able to predict properties of the molecules, we add a property-prediction loss to the training objective. During training, the model is tasked with predicting both the next token in

the sequence and the properties of the current unfinished molecule. During sampling, we generate molecules conditioned on the desired properties with classifier-free guidance. Then, we mask out the properties (making them fully missing) and reprocess the molecule until we reach the end-of-sequence (EOS) token. At this point, we extract the predicted property of this molecule.

The architecture with the described loss functions is illustrated in Figure 1. The methodology for generating molecules using classifier-free guidance and self-criticism is depicted in Figure 2.

### 3.6 RANDOM GUIDANCE FOR EXTREME CONDITIONING

Regular guidance can be problematic for some extreme (out-of-distribution) conditioning values, resulting in poor generative efficiency (% of valid, unique, and novel molecules) and conditioning fidelity. However, guidance can still be beneficial to some extreme conditioning values.

To improve generative performance on extreme conditioning values, we propose to randomly sample a guidance $w \sim \mathcal{U}(-0.5, 2)$ for each sample, ensuring high diversity through a mix of low and high guidance. Then, using self-criticism, our method selects the best-out-of-$k$ molecule from the molecules generated at different guidance levels, indirectly allowing the model to determine by itself which guidance is best for each sample. This is effectively a way to balance exploration and exploitation (higher guidance means less diversity and better property-alignment, while lower guidance means more diversity and less property-alignment).

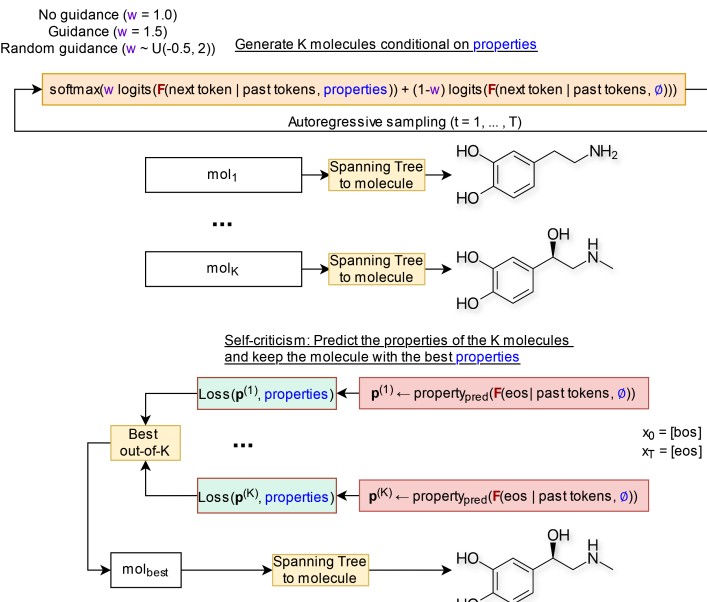

Figure 2: Generation and self-prediction using **STGG+** . We autoregressively generate $K$ molecules conditional on desired properties using classifier-free guidance. The unconditional model predicts the properties of the $K$ molecules and the molecule assumed closest to the desired properties is returned.

## 4 EXPERIMENTS

We run four sets of experiments. First, we demonstrate that, after conditioning on in-distribution properties, our model can recover molecules close to those in the test set, achieving performance on par with state-of-the-art unconditional models. Second, we show that our model can generate molecules conditioned on properties from the test set with high fidelity on the specified properties. Third, we illustrate that our model can generate highly efficient (high % of novel, unique, and valid) molecules with high fidelity on out-of-distribution (OoD) properties. Fourth, we show that our model can produce molecules that maximize a reward function, achieving similar or better performance compared to online learning methods using offline learning. Finally, as a harder case, we show that

our model can generate high fidelity molecules conditioned on out-of-distribution (OoD) properties on a small dataset of large molecules.

See Appendix A.3 for details on the datasets used, Appendix A.4 for more information on the hyperparameters, Appendix A.6 for property prediction performance metrics of the self-critic, and Appendix A.8 for an ablation on OOD properties for Zinc. Note that we rely on the following software: PyTorch (Paszke et al., 2019), Molecular Sets (MOSES) (Polykovskiy et al., 2020) and RDKit: Open-source cheminformatics (Landrum et al., 2024). Unless otherwise specified, we use RDKit to evaluate the properties of the generated molecules.

## 4.1 UNCONDITIONAL GENERATION

We train our model on QM9 (Ramakrishnan et al., 2014) and Zinc250K (Sterling & Irwin, 2015) using the molecule weight, logP, and Quantitative Estimate of Druglikeness (QED) (Bickerton et al., 2012) as properties. We test the similarity in the distribution of unconditionally generated molecules (masking the properties). We also test the same metrics on conditionally generated molecules, conditioned on properties from the test set. We use the same train, valid, and test splits as Jo et al. (2022). We compare to the strong recent baselines reported in Jang et al. (2023) which are: EDP-GNN (Niu et al., 2020), GraphAF (Shi et al., 2020), GraphDF (Luo et al., 2021), GDSS (Jo et al., 2022), DiGress (Vignac et al., 2022), DruM (Jo et al., 2023), GraphARM (Kong et al., 2023), GEEL (Jang et al., 2023), CharRNN (Segler et al., 2018), CG-VAE (Liu et al., 2018), MoFlow (Zang & Wang, 2020), and STGG (Ahn et al., 2021). The metrics are % Valid, % unique, % novel, Fréchet ChemNet Distance (FCD) (Preuer et al., 2018), scaffold similarity (Scaf.), similarity to nearest neighbor (SNN), and fragment similarity (Frag.).

**Results** The top performing methods (STGG, GEEL, STGG+) are shown in Table 1; they have similar performance. The full experiments can be found in Appendix A.7.

Table 1: Unconditional molecular graph generation performance.

| Method | Valid (%) (↑) | Unique (%) (↑) | Novel (%) (↑) | FCD (↓) | Scaf. (↑) | SNN (↑) | Frag. (↑) |
|---|---|---|---|---|---|---|---|
| | | | QM9 | | | | |
| GEEL | **100.0** | 96.08 | 22.30 | **0.089** | 0.9386 | 0.5161 | 0.9891 |
| STGG | **100.0** | 96.76 | 72.73 | 0.585 | **0.9416** | **0.9998** | **0.9984** |
| STGG+ | **100.0** | **97.17** | **74.41** | **0.089** | 0.9265 | 0.5179 | 0.9877 |
| | | | Zinc250K | | | | |
| GEEL | 99.31 | 99.97 | 99.89 | 0.401 | 0.5565 | 0.4473 | 0.9920 |
| STGG | **100.0** | **99.99** | 99.89 | **0.278** | **0.7192** | **0.4664** | **0.9932** |
| STGG+ | **100.0** | **99.99** | **99.94** | 0.395 | 0.5657 | 0.4316 | 0.9925 |

## 4.2 CONDITIONAL GENERATION

We follow the same protocol as Liu et al. (2024). We train our model on HIV, BACE, and BBBP (Wu et al., 2018). We use the same train, valid, and test splits as Liu et al. (2024). Each dataset has a experimental categorical property related to HIV virus replication inhibition (HIV), blood-brain barrier permeability (BBBP), or human $\beta$-secretase 1 inhibition (BACE), respectively, and two continuous properties: synthetic accessibility (SAS) (Ertl & Schuffenhauer, 2009) and complexity scores (SCS) (Coley et al., 2018). We evaluate the models using metrics on distribution and fidelity of conditioning after generating molecules conditional on properties from the test set. The condition control metrics are the Mean Absolute Error (MAE) of SAS (evaluated by RDKit) and accuracy of the categorical property (evaluated by a Random Forest (Breiman, 2001) predictor using the Morgan Fingerprint (Morgan, 1965; Gao et al., 2022)). The distribution metrics are Validity, atom coverage in the largest connected graph (how many unique atom types are produced in the generated samples), internal diversity (average pairwise similarity of generated molecules), fragment-based similarity (Degen et al., 2008), Fréchet ChemNet Distance (FCD) (Preuer et al., 2018). We consider any atom coverage above the test set coverage to indicate good coverage. The Property accuracy metric depends on a RandomForest classifier, thus we consider any accuracy equal or above the test set to indicate good condition control.

Notably, the Fréchet Distance is one of the most popular and meaningful distance in generative models as it correlates well with both quality and diversity (Heusel et al., 2017); it corresponds to the

Wasserstein distance on the hidden space of neural networks assuming normality. It has been used in many domains: images (Heusel et al., 2017), audio (Kilgour et al., 2018), videos (Unterthiner et al., 2019), and molecules (Preuer et al., 2018).

Following Liu et al. (2024), we compare our method to strong recent baselines: GraphGA (Jensen, 2019), MARS (Xie et al., 2021), LSTM on SMILES with Hill Climbing (LSTM-HC) (Hochreiter & Schmidhuber, 1997; Brown et al., 2019), and powerful graph diffusion models: DiGress (Vignac et al., 2022), GDSS Jo et al. (2022), and MOOD (Lee et al., 2023), and Graph DiT (Liu et al., 2024).

**Results** The experiments are shown in Table 2 (for the full table with more baselines, see Appendix A.9). We find that STGG+ obtains near-perfect validity, coverage consistently higher than the test set, high diversity, and high test-set similarity. Notably, we attain the best FCD; in fact, we are the only method that matches the training data's performance, indicating that we have reached the performance cap. Regarding condition control, we achieve the best MAE on BACE and HIV, and the second-best on BBBP (very close to Graph DiT). We also obtain better performance than base STGG (with random masking and the extra symbol for compounds) on FCD and MAE, which shows that our improvements lead to lower distance in distribution and better property conditioning.

Table 2: Conditional generation of 10K molecular compounds on HIV, BBBP, and BACE.

| Tasks | Metric | Validity ↑ | Distribution Learning | | | | Condition Control | |
|---|---|---|---|---|---|---|---|---|
| | | | Coverage* ↑ | Diversity ↑ | Similarity ↑ | FCD ↓ | MAE ↓ | Accuracy * ↑ |
| | Model \ Property | | | | | | SAS | BACE, BBBP, or HIV |
| SAS & BACE | MOOD | 1.00 | 8/8 | 0.89 | 0.26 | 44.24 | 1.89 | 0.51 |
| | Graph GA | 1.00 | 8/8 | 0.86 | 0.98 | 7.41 | 0.96 | 0.47 |
| | Graph DiT | 0.87 | 8/8 | 0.82 | 0.88 | 7.05 | 0.40 | 0.91 |
| | STGG** | 1.00 | 8/8 | 0.82 | 0.98 | 3.82 | 0.45 | 0.95 |
| | **STGG+** ($k=1$) | 1.00 | 8/8 | 0.83 | 0.98 | 3.80 | 0.24 | 0.91 |
| | **STGG+** ($k=5$) | 1.00 | 8/8 | 0.83 | 0.98 | 3.80 | 0.18 | 0.93 |
| | **Test data** | 1.00 | **7/8**$^*$ | 0.82 | 1.00 | 0.00 | 0.00$^\dagger$ | **0.82**$^*$ |
| SAS & BBBP | MOOD | 0.80 | 9/10 | 0.93 | 0.17 | 34.25 | 2.03 | 0.49 |
| | Graph GA | 1.00 | 9/10 | 0.90 | 0.95 | 10.17 | 1.21 | 0.30 |
| | Graph DiT | 0.85 | 9/10 | 0.89 | 0.93 | 11.85 | 0.36 | 0.94 |
| | STGG** | 1.00 | 9/10 | 0.89 | 0.92 | 11.74 | 0.98 | 0.75 |
| | **STGG+** ($k=1$) | 1.00 | 10/10 | 0.89 | 0.94 | 9.86 | 0.47 | 0.87 |
| | **STGG+** ($k=5$) | 1.00 | 9/10 | 0.89 | 0.94 | 10.10 | 0.38 | 0.90 |
| | **Test data** | 1.00 | **10/10**$^*$ | 0.88 | 1.00 | 0.00 | 0.02$^\dagger$ | **0.81**$^*$ |
| SAS & HIV | MOOD | 0.29 | 29/29 | 0.93 | 0.14 | 32.35 | 2.31 | 0.51 |
| | Graph GA | 1.00 | 28/29 | 0.90 | 0.97 | 4.44 | 0.98 | 0.60 |
| | Graph DiT | 0.77 | 28/29 | 0.90 | 0.96 | 6.02 | 0.31 | 0.98 |
| | STGG** | 1.00 | 27/29 | 0.90 | 0.96 | 4.56 | 0.44 | 0.95 |
| | **STGG+** ($k=1$) | 1.00 | 27/29 | 0.90 | 0.97 | 4.08 | 0.31 | 0.88 |
| | **STGG+** ($k=5$) | 1.00 | 24/29 | 0.90 | 0.97 | 4.32 | 0.23 | 0.91 |
| | **Test data** | 1.00 | **21/29**$^*$ | 0.90 | 1.00 | 0.07 | 0.02$^\dagger$ | **0.73**$^*$ |

$^*$The classifier from Liu et al. (2024) (used in the last column) has limited accuracy on the test set; thus, any *Property Acc.* above the **test data accuracy** is not indicative of better quality. Similarly, atom coverage is not 100% on test data; thus, any coverage above the **test set coverage** does not indicate better performance.
$^{**}$STGG with categorical embedding, missing indicators, random masking, and extra symbol for compounds.
$^\dagger$The dataset properties are rounded to two decimals hence MAE is not exactly zero.

### 4.3 OUT-OF-DISTRIBUTION CONDITIONAL GENERATION

We follow the same protocol as Kwon et al. (2023). Our model is trained on Zinc250K (Sterling & Irwin, 2015) using exact molecule weight, logP, and Quantitative Estimate of Druglikeness (QED) (Bickerton et al., 2012) as properties. For evaluation, we generate 2K candidate molecules and calculate two metrics: 1) generative efficiency, defined as the probability that the following three conditions are satisfied: validity, uniqueness (not a duplicate), and novelty (not in train data)), and 2)

the Minimum Mean Absolute Error (MinMAE) between the generated and conditioned properties (at $\pm 4$ standard-deviation). Note that for QED, the high condition value is at an impossible value of 1.2861 (the possible range is 0 to 0.948). Conditioning on impossible values is not ideal, but we must follow Kwon et al. (2023) protocol and its useful to test how the model behave in erroneous scenarios. We use the same train, valid, and test splits as Jo et al. (2022). Following Shao et al. (2020), we compare our model to vanilla VAE with k-annealing (BaseVAE) (Kingma & Welling, 2013; Bowman et al., 2015), ControlVAE (Shao et al., 2020), and various single-decoder (SD) and multi-decoders (MD) methods proposed by Shao et al. (2020).

**Results** The experiments are shown in Table 3 (see Table 12 for the top-100 MAE). We see that base STGG (with random masking) reaches the best generative efficiency (% of valid, novel, and unique molecules), but performs much worse than **STGG+** in terms of property conditioning. Our method sacrifices a small amount of generative efficiency (when compared to base STGG) in order to obtain much better property-conditioning; we see that our method generally obtains the smallest MAE. However, while the model performs optimally when using random guidance, it struggles with high guidance values when generating molecules for the impossible QED value of 1.2861. Additionally, we observe that the model performs worse with the best-of-5 when generating molecules high logP, suggesting that the property predictor of STGG+ makes incorrect predictions for out-of-distribution high logP values.

Table 3: Out-of-distribution ($\mu \pm 4\sigma$) property-conditional generation of 2K molecules on Zinc250K. Generative efficiency (% of valid, novel, and unique molecules) and Minimum MAE (MinMAE).

| | Generative Efficiency | | | Properties - MinMAE | | | | | |
| | molWt | logP | QED | molWt | | logP | | QED | |
| Condition | | | | 84 | 580 | -3.2810 | 8.1940 | 0.1778 | 1.2861* |
| $MD$ | 0.49 | 0.42 | 0.47 | 9.8e−2 | 1.7e−1 | 2.0e−2 | 3.0e−4 | 1.5e−3 | 1.0e−1 |
| $MD_{dif}$ | 0.46 | 0.43 | 0.47 | 7.4e−3 | 4.7e−2 | 3.0e−4 | 5.1e−3 | 2.0e−4 | 2.6e−2 |
| $MD_{dif,col}$ | 0.46 | 0.54 | 0.44 | 1.1e−1 | 6.2e−2 | 1.3e−3 | 5.0e−4 | 6.0e−4 | 8.6e−2 |
| STGG** | 0.99 | 0.99 | 0.99 | 5.8e−2 | 7.5e−2 | 7.9e−3 | 1.9e−1 | 1.5e−2 | 8.0e−4 |
| **STGG+** ($k = 1$) | 0.82 | 0.82 | 0.54 | 8.6e−3 | 9.1e−3 | 1.0e−4 | 1.6e−3 | 1.0e−5 | 5.1e−1 |
| **STGG+** ($k = 5$) | 0.88 | 0.74 | 0.50 | 1.1e−3 | 1.7e−2 | 1.0e−4 | 1.6e+0 | 1.0e−4 | 5.2e−1 |
| **STGG+** ($w \sim \mathcal{U}(-0.5, 2), k = 1$) | 0.94 | 0.92 | 0.82 | 2.1e−2 | 2.4e−2 | 1.0e−4 | 7.0e−4 | 7.0e−6 | 5.8e−3 |
| **STGG+** ($w \sim \mathcal{U}(-0.5, 2), k = 5$) | 0.90 | 0.77 | 0.79 | 1.0e−3 | 6.1e−3 | 2.0e−7 | 2.8e−2 | 1.0e−4 | 1.2e−3 |
| **Train data** (closest sample) | - | - | - | 5.7e+1 | 7.3e+1 | 1.5e−1 | 2.0e−3 | 1.8e−2 | 8.2e−4 |

\*The value is improper; we condition on 1.2861 but calculate the MAE with respect to the maximum QED (0.948).
\*\*STGG with missing indicators, and random masking.

## 4.4 REWARD MAXIMIZATION

Jain et al. (2023) train reinforcement learning (RL) and GFlowNet (Bengio et al., 2023) agents to solve a task based on the QM9 (Ramakrishnan et al., 2014) dataset. They seek to produce QM9-like molecules which maximize a reward composed of four properties: HOMO-LUMO gap (Griffith & Orgel, 1957), SAS (Ertl & Schuffenhauer, 2009), QED (Bickerton et al., 2012), and molecular weight. This reward is maximized when the HOMO-LUMO gap is as large as possible, and SAS, QED, and weight are 2.5, 1.0, and 105, respectively. We compare to Envelope QL (Yang et al., 2019), MOReinforce (Lin et al., 2022), MOA2C (Mnih et al., 2016), GFlowNet (MOGFN-PC) (Bengio et al., 2023). The HOMO-LUMO gap is evaluated with MXMNet (Zhang et al., 2020).

Instead of giving the reward to our model, we train a STGG+ model conditioned on the four properties. Since the HOMO-LUMO gap needs to be maximized there is no appropriate conditioning value. We arbitrarily set it to 0.5, which corresponds to approximately five standard deviations (a limitation of our conditioning method, as we cannot maximize a property, only set a fixed value). The other properties are set to their optimal values: 2.5, 1.0, and 105.

**Results** Our experiments are shown in Table 4. Our approach yields slightly better molecules in terms of reward and diversity compared to online methods, using around 11.5% of the molecules. This makes our approach significantly more efficient. However, it is important to note that solving this task with online methods is a steep hill and can be considered more difficult.

## 4.5 HARD: SMALL DATASET OF LARGE MOLECULES (CHROMOPHORE DB)

As a more challenging example, we explore the generation of molecules with out-of-distribution properties on Chromophore DB (Joung et al., 2020), a small dataset of around 6K molecules with an

Table 4: Reward maximization on QM9.

| | Type | Data | Reward (↑) | Diversity (↑) |
|---|---|---|---|---|
| Envelope QL | Online | 1M molecules | 0.65 | 0.85 |
| MOGFN-PC | | | 0.76 | 0.93 |
| STGG** | Offline | QM9 ($\sim$115K molecules) | 0.73 | 0.10 |
| **STGG+** ($k = 1$) | | | 0.78 | 0.76 |
| **STGG+** ($k = 100$) | | | 0.77 | 0.98 |

**STGG with missing indicators, and random masking.

average of 35 atoms per molecule (compared to 23 atoms for Zinc250K and 9 atoms for QM9). To make the problem more realistic, we only sample 100 molecules (in the real world, chemists would decide which of those 100 molecules to synthesize based on their expert knowledge). We want to know if one of those 100 molecules has the desired out-of-distribution properties.

Given the small size of the dataset, it might be useful to first pre-train on a large set of small molecules (Zinc250K) and then fine-tune on the smaller dataset of large molecules (Chromophore DB). We try training with this strategy (pre-train and fine-tune) in addition to only training on Chromophore DB.

**Results** The experiments are shown in Table 5 (see Table 13 for the top-100 MAE). We find that pre-training on Zinc250K generally improves performance (Generative Efficiency and MinMAE) over training only on Chromophore DB. For most properties, random guidance with filtering ($k > 1$) leads to the closest properties. However, for high logP, we obtain better property fidelity with no filtering ($k = 1$), indicating that the model struggles with property prediction on large out-of-distribution logP values.

Table 5: Out-of-distribution ($\mu \pm 4\sigma$) property-conditional generation of 100 molecules on Chromophore DB. Generative Efficiency (% of valid, novel, and unique molecules) and Minimum MAE (MinMAE). We removed the low molWt and QED which are both impossible negative values.

| | Generative Efficiency | | | | Properties - MinMAE | | | |
|---|---|---|---|---|---|---|---|---|
| | molWt | logP | | QED | molWt | logP | | QED |
| Condition | 1538.00 | -13.63 | 28.69 | 1.24* | 1538.00 | -13.63 | 28.69 | 1.24* |
| Trained on Chromophore DB (1000 epochs) | | | | | | | | |
| **STGG+** ($k = 1$) | 0.97 | 0.33 | 0.98 | 0.59 | 9.02 | 3.30 | 0.03 | 0.30 |
| **STGG+** ($k = 100$) | 0.88 | 0.25 | 0.82 | 0.81 | 5.24 | 6.02 | 8.02 | 0.25 |
| **STGG+** ($w \sim \mathcal{U}(-0.5, 2), k = 1$) | 0.91 | 0.71 | 0.92 | 0.75 | 0.41 | 8.10 | 0.12 | 0.05 |
| **STGG+** ($w \sim \mathcal{U}(-0.5, 2), k = 100$) | 0.89 | 0.71 | 0.94 | 0.83 | 0.74 | 0.89 | 7.03 | 0.01 |
| Pre-trained on Zinc250K (50 epochs) and fine-tuned on Chromophore DB (100 epochs) | | | | | | | | |
| **STGG+** ($k = 1$) | 0.99 | 0.96 | 0.99 | 0.98 | 0.94 | 0.38 | 0.41 | 0.15 |
| **STGG+** ($k = 100$) | 1.00 | 0.96 | 0.93 | 1.00 | 2.37 | 0.35 | 0.42 | 0.09 |
| **STGG+** ($w \sim \mathcal{U}(-0.5, 2), k = 1$) | 1.00 | 0.95 | 0.97 | 1.00 | 0.47 | 0.66 | 0.01 | 0.02 |
| **STGG+** ($w \sim \mathcal{U}(-0.5, 2), k = 100$) | 1.00 | 0.92 | 0.98 | 0.99 | 13.19 | 0.45 | 0.18 | 0.01 |
| **Train data** (closest sample) | - | - | | - | 1.40 | 9.62 | 0.17 | 0.01 |

*The value of 1.24 is improper; we calculate the MAE with respect to the maximum QED (0.948).

## 5 CONCLUSION

In this paper, we demonstrated that with specific techniques, optimization, and architectural improvements, spanning tree-based graph generation (STGG) can be leveraged to generate high-quality and diverse molecules conditioned on both *in-distribution* and *out-of-distribution* properties. Our method achieves equal or superior performance on validty, novelty, uniqueness, closeness in distribution, and conditioning fidelity compared to competing approaches while being extremely efficient and fast. Using fewer molecules than required by online methods (RL/GFlowNet), we also obtain high multi-property-reward molecules in a one-shot manner from a pre-trained model.

While our method generates molecules with relatively good accuracy concerning the desired properties, it is still not perfect and can produce incorrect molecules, especially in out-of-distribution scenarios, which is a challenging task. Additionally, the property predictor of our approach may not be as optimal as property predictors engineered explicitly for this task, meaning our method may not always select the best molecules out of $k$ choices, particularly in out-of-distribution scenarios; we found this to be the case for large out-of-distribution logP conditioning values. Currently, the method does not account for stereoisomers, although some properties can be dependent on stereoisomers.

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

# A  APPENDIX

## A.1  1D VS 2D REPRESENTATIONS

There are many ways to represent molecules in the context of molecular generation. The most popular methods are autoregressive models on 1D strings and diffusion (Song & Ermon, 2019; Ho et al., 2020; Song et al., 2020) models on 2D graphs. We highlight the main distinction between the two representations below in the context.

Let $D$ be the size of the training dataset, $n$ be the number of atoms in a given molecule, $d$ is the embedding size, and $b$ is the number of bond types.

Diffusion models on 2D graphs:

- $G = (X, A)$ where the vertices $X$ contains the list of atoms (size: $[n, d]$) and $A$ is the adjacency matrix of the edges (size: $[n, n, b]$) for each bond type.
- $A$ is an extremely sparse matrix with many zero elements
- Input space is $\mathcal{O}(nd + bn^2)$; unless using low-rank projections, the number of parameters must scale proportionally to this amount
- Typically use diffusion models (or related methods) given the large number of steps it would take to generate $X$ and $A$ autoregressively
- Equivariant Graph Neural Networks (E-GNNs) are generally used to ensure a unique representation for a given molecule
- Although it has a single representation per molecule, multiple random noises per graph are needed due to diffusion; thus, sample complexity is $\mathcal{O}(Dn_{noise})$

Autoregressive models on 1D strings:

- $X$ (size: $[L, d]$) is a string containing the molecule where $L$ is proportional to $n$
- The string starts from a random atom and traverses the 2D molecular graph
- Input space is $\mathcal{O}(nd)$; this makes it efficient to process
- Typically use autoregressive models (e.g., Transformers) as it scales well
- We can either 1) fix the ordering in some way to make representation unique, or 2) use random orderings as data augmentation with a non-unique representation for a given molecule; thus, sample complexity is $\mathcal{O}(Dn_{augments})$

As can be seen, both methods have similar sample complexity, but 1D strings are much more compressed representations, leading to less space and parameters and, thus, increased potential for scalability. Furthermore, recent results show that 1D strings are as competitive as 2D molecular graph methods for unconditional molecule generation Jang et al. (2023); Fang et al. (2023) and property prediction Yüksel et al. (2023). In the end, both representations contain as much information. Thus, the choice is a matter of preference. 1D strings are easier to scale and can make good use of the power of Transformers; hence, we focus on this type of representation.

## A.2  SPANNING TREE COMPARED TO OTHER 1D STRING-BASED REPRESENTATIONS

The most popular choice of string-based representation is SMILES (Weininger, 1988). SMILES is extremely versatile, allowing the representation of any molecule. However, for the purpose of generative models, trying to generate SMILES strings directly can quickly lead to many invalid molecules. Graph-based diffusion methods encounter the same issue. Recently, methods have been created to prevent the creation of invalid molecules: Spanning Tree (Ahn et al., 2021) and SELFIES (Krenn et al., 2022). Below, we describe in detail the differences between all three methods.

SMILES:

- Massive vocabulary allows the representation of every aspect of molecules
- There are many ways of representing a single molecule
- Begin-branch token "(" to deviate from the main path and close branch token ")"

- Pointer token $i$ to indicate both the beginning and end of rings

SELFIES:

- Restricted SMILES vocabulary
- Prevent invalid molecules through a carefully designed context-free grammar:
    - Atoms and bonds are combined into single tokens (with other aspects such as charge and number of hydrogen atoms) so that we cannot have an atom without a bond and a bond without an atom
    - Hard-designed rules for maximum valencies of specific elements (slightly more permissible than octet rule, but cannot handle every case)
    - Keep track of valencies; ignore future tokens in the current branch if there is not enough valency left and reduce bond order if needed
    - There is an open-branch token Branch-$i$ and close-ring token ring-$i$ where $i$ specifies the number of future tokens in the branch and how many backward steps (in tokens) are needed to reach the ring closure; this ensures that all branches and tokens are not left opened

Spanning-Tree:

- Restricted SMILES vocabulary
- Begin and end branch tokens, with ring start and $i$-th ring end tokens
- Similarity-based output layer to determine the probabilities of ring ends and input embedding injection for how many rings are opened
- Prevent invalid molecules through masking of tokens before softmax:
    - Masking of invalid tokens due to impossible valencies (atoms, ring-start, and branch-start when not enough valency is left) based on the valencies of the training data
    - Masking of invalid next tokens (atom after atom, bond after bond, ring-$i$ end when there are less than $i$ ring start tokens)
    - Force branch ending through masking when getting too close to maximum sequence length to prevent unfinished molecules (new)

As can be seen, SMILES has such a large vocabulary that each molecule can be represented in completely different ways, and its main problem for generative models is that many token choices lead to invalid molecules (e.g., two bonds, incorrect valencies, unfinished branches, or rings, etc.).

SELFIES prevents invalid molecules through its smart, context-free grammar. Although powerful, this grammar risks making the job of the generative model more difficult as it requires the model to plan in advance how many tokens will be in each branch and also to count backward to determine how many backward steps it must take to reach the ring-end.

On the other hand, Spanning-tree use clever masking of incorrect tokens to prevent invalid molecules and doing so does not require the model to do significant planning-in-advance and counting when selecting the next token (including the $i$-th ring end tokens which require no counting due to the similarity-based prediction).

### A.3 DATASETS DETAILS AND CANONICALIZATION

QM9 (Ramakrishnan et al., 2014) has 21 atom tokens: CH3, C, O, CH2, CH, NH, N, N-, NH+, OH, NH2, F, NH3+, O-, NH2+, N+, C-, CH-, NH3, OH2, CH4. The maximum length is 37. The dataset has 133886 molecules with around 10% of the molecules in the test set and 5% in the validation set.

Zinc250K (Sterling & Irwin, 2015) has 34 atom tokens: CH3, C, CH, N, S, CH2, O, NH, NH+, NH2, NH2+, NH3+, OH, Cl, O-, N-, F, Br, N+, S-, I, SH, P, NH-, O+, OH+, S+, CH2-, CH-, SH+, PH, PH+, P+, PH2. The maximum length is 136. The dataset has 133886 molecules with around 10% of the molecules in the test set and 5% in the validation set.

BBBP (Wu et al., 2018) has 31 atom tokens: CH, C, F, CH2, N, S, CH3, O, OH, NH2, Cl, NH, OH2, Br, O-, N+, Na, Cl-, H+, C-, Na+, NH+, NH3+, Br-, P, N-, SH, CH2-, CH-, I, B. The maximum

length is 186. The dataset has 862 molecules with around 20% of the molecules in the test set and 20% in the validation set.

BACE (Wu et al., 2018) has 20 atom tokens: F, C, N, CH, NH+, CH2, NH2, O, Cl, S, CH3, NH, OH, NH2+, Br, O-, NH3+, N+, N-, I. The maximum length is 161. The dataset has 1332 molecules with around 20% of the molecules in the test set and 20% in the validation set.

HIV (Wu et al., 2018) has 76 atom tokens: CH3, C, O, CH2, N, NH2, CH, N+, NH2+, I, NH, Br, Se, OH, S, O-, Br-, SH, Cl, I-, S+, Zn-2, OH+, N-, NaH, PH, Ir-3, Cl-, NH3, F, P, BrH, C-, Co-2, Cu-4, As, B-2, Sn, ClH, Rh-4, O+, S-, Pt, Fe-2, B, U+2, Pd-2, Fe-3, Pt-2, Pt+2, Si, P+, IH2, Fe, SiH, Cl+3, Ge, NH+, Zr, K+, AlH3-, IH, KH, Mn+, Fe-4, Cu-3, Ni-4, LiH, Co-3, Pd-3, Fe+2, Ga-3, CH2-, U, Mn, Co-4. The maximum length is 193. The dataset has 2372 molecules with around 20% of the molecules in the test set and 20% in the validation set.

Chromophore DB (Joung et al., 2020) has 46 atom tokens: CH, C, N, CH3, CH2, O, N+, B-, F, S, OH, NH, Cl, NH2, P, O+, Si, O-, Se, C+, B, Br, I, NH+, NH2+, N-, S+, SiH, C-, Na, Sn, NH3+, S-, Si-, P-, Cl+3, I-, BH3-, P+, BH, CH4, NH-, SH, Ge, Te, Na+. The maximum length is 511. The dataset has 6810 molecules with around 5% of the molecules in the test set and 5% in the validation set.

Note that we base the maximum length on the largest SMILES string after being transformed with the Spanning tree tokenizer.

### A.3.1 CANONICALIZATION

Similar to STGG (Ahn et al., 2021), we use explicit Hydrogen atoms (with no implicit Hydrogen atom) in the tokens. This is an abitrary choice. After generation, we always transform back to canonical SMILES using RDKit (Landrum et al., 2024). Note that RDKit may change the number of Hydrogen atoms based on its own rule-set. All our molecule figures are based on RDKit so they reflect the molecules after SMILES canonicalization by RDKit.

Here is an example below.

Training SMILES: C[C@@]12C=CC(=O)C=C1CC[C@@H]1[C@@H]3CC[C@](O)(C(=O)COP(=O)([O-])[O-])[C@]3(C)C[C@@H](O)[C@H]12

STGG tokenized: [bos]C-C[bor][bor]-C=C-C(=O)(-C=C(-[eor0])(-C-C-CH[bor]-CH[bor]-C-C-C(-O)(-C(=O)(-C-O-P(=O)(-O-)(-O-)))(-C(-[eor3])(-C)(-C-CH(-O)-CH(-[eor1])(-[eor2])))))[eos]

Canonical SMILES: CC12C=CC(=O)C=C1CCC1C2C(O)CC2(C)C1CCC2(O)C(=O)COP(=O)([O-])[O-]

### A.3.2 ANY-PROPERTY MASKING

For any-property masking, we show an example below. Assume that there are $T = 5$ properties. Each time we sample a training molecule, we choose a random number $t$ of properties to mask uniformly between 0 and $T = 5$. Assuming that $t = 3$, we create the masking vector: [1, 1, 1, 0, 0, 0]. Then, we randomly shuffle the masking vector, leading to: [1, 0, 0, 1, 0, 1]. Then, we mask the properties with a masking value of 1.

### A.4 HYPERPARAMETERS

The original STGG (Ahn et al., 2021) used the AdamW optimizer (Loshchilov & Hutter, 2017; Kingma & Ba, 2014) with $\beta_1 = 0.9$, $\beta_2 = 0.999$, no weight decay, a fixed learning rate of 2e-4 and batch-size 128. The Transformer architecture had 3 layers, dropout 0.1, 8 attention heads, and embedding size 1024. They processed only one property with a single linear layer.

**STGG+** uses the AdamW optimizer with $\beta_1 = 0.9$, $\beta_2 = 0.95$, and weight decay 0.1. The Transformer architecture has 3 layers, no dropout, 16 attention heads, SwiGLU (Hendrycks & Gimpel, 2016; Shazeer, 2020) with expansion scale of 2, no bias term (Chowdhery et al., 2023), Flash Attention (Dao et al., 2022; Dao, 2023), RMSNorm (Zhang & Sennrich, 2019), Rotary embeddings (Su et al., 2024), residual-path weight initialization (Radford et al., 2019).

For QM9 (Ramakrishnan et al., 2014), we train for 50 epochs with batch size 512, learning rate 1e-3, max length 150. For Zinc250K (Sterling & Irwin, 2015), we train for 50 epochs with batch size 512, learning rate 1e-3, max length 250. For HIV, BACE, and BBBP (Wu et al., 2018), we train for 10K epochs (same as done by Liu et al. (2024)), since these are small datasets, with batch size 128, learning rate 2.5e-4, max length 300.

For Chromophore DB (Joung et al., 2020), we train for 1000 epochs with batch size 128, learning rate 2.5e-4, max length 600. For the pre-training on Zinc250K and fine-tuning on Chromophore-DB: we pre-train with batch size 512, learning rate 1e-3, and max length 600 for 50 epochs and fine-tune with batch size 128, learning rate 2.5e-4, and max length 600 for 100 epochs.

We generally use 1 or 2 A-100 GPUs to train the models. Training takes a few hours. Note that we use a higher max length than the data max length (generally around 25-50%) to ensure that we can adequately generate molecules with out-of-distribution properties that could be bigger than usual.

For pretraining and then fine-tune, there are two ways to preprocess the properties: we can either standardize them with respect to the pre-training or the fine-tuning datasets. Standardizing with respect to the pre-training dataset can lead to extreme values in the fine-tuning (e.g., 4 standard deviation in Chromophore's MolWt is 15 standard-deviation in Zinc250K's MolWt). Hereby, to reduce the gap between pre-trained and fine-tuned conditioning values, we preprocess the properties by standardizing with respect to the fine-tune dataset properties during both pre-training and fine-tuning.

## A.5 Alternative architectures considered

In this work, we enhance the Transformer architecture used by Ahn et al. (2021) using recent developments in Large Language Models (LLMs). Although powerful, the Transformer architecture with self-attention (Vaswani et al., 2017) is quadratic in context length, which means that the time and memory increase significantly when dealing with long-context length.

In addition to improvements on Transformer, new architectures such as Mamba (Gu & Dao, 2023), Hyena, (Poli et al., 2023) or RWKV (Peng et al., 2023) have appeared, which are sub-quadratic with respect to context-length, allowing them to handle long-context length better. We initially considered some of these architectures to improve inference speed. However, it is hard to synthesize and manufacture molecules of substantial sizes. Thus, the context length is generally quite limited (e.g., the largest molecule on Chromophore has 511 tokens, while modern LLMs have a context length of at least 4096). As long as the context length is less or equal to 2048, FlashAttention (Dao et al., 2022) is fast enough that there is no inference speed benefit for using Mamba (Gu & Dao, 2023).

## A.6 Property prediction

Table 6: Property prediction on the test set using **STGG+** or a Random Forest (Breiman, 2001) predictor/classifier using the Morgan Fingerprint (Morgan, 1965) as done by Gao et al. (2022).

| Task | Method | Accuracy | Mean squared error (MSE) | | | | | |
| | | HIV | QED | MolWt | logP | SAS | SCS | Gap |
|---|---|---|---|---|---|---|---|---|
| QM9 | STGG+ | - | 0.0010 | 0.0018 | 0.0012 | - | - | - |
| QM9 | Random Forest | - | 0.2665 | 0.6124 | 0.2014 | - | - | - |
| Zinc250K | STGG+ | - | 0.0008 | 0.0005 | 0.0005 | - | - | - |
| Zinc250K | Random Forest | - | 0.4077 | 0.4209 | 0.3907 | - | - | - |
| HIV | STGG+ | 0.8463 | - | - | - | 0.0268 | 0.0216 | - |
| HIV | Random Forest | 0.7263 | - | - | - | 0.3605 | 0.4672 | - |
| BACE | STGG+ | 0.9551 | - | - | - | 0.0126 | 0.0070 | - |
| BACE | Random Forest | 0.8165 | - | - | - | 0.1773 | 0.3948 | - |
| BBBP | STGG+ | 0.8743 | - | - | - | 0.0354 | 0.0314 | - |
| BBBP | Random Forest | 0.8057 | - | - | - | 0.3152 | 0.4740 | - |
| QM9 (Reward) | STGG+ | - | - | 0.0009 | 0.0005 | 0.0021 | - | 0.0032 |
| QM9 (Reward) | Random Forest | - | - | 0.6122 | 0.2015 | 0.2114 | - | 0.1459 |

## A.7 UNCONDITIONAL GENERATION

Table 7: Molecular graph generation performance on QM9.

| Method | Valid (%) (↑) | Unique (%) (↑) | Novel (%) (↑) | FCD (↓) | Scaf. (↑) | SNN (↑) | Frag. (↑) |
|---|---|---|---|---|---|---|---|
| *Domain-agnostic graph generative models* | | | | | | | |
| EDP-GNN | 47.52 | **99.25** | 86.58 | 2.680 | 0.3270 | 0.5265 | 0.8313 |
| GraphAF | 74.43 | 88.64 | 86.59 | 5.625 | 0.3046 | 0.4040 | 0.8319 |
| GraphDF | 93.88 | 98.58 | **98.54** | 10.928 | 0.0978 | 0.2948 | 0.4370 |
| GDSS | 95.72 | 98.46 | 86.27 | 2.900 | 0.6983 | 0.3951 | 0.9224 |
| DiGress | 98.19 | 96.67 | 25.58 | 0.095 | 0.9353 | 0.5263 | 0.0023 |
| DruM | 99.69 | 96.90 | 24.15 | 0.108 | **0.9449** | 0.5272 | 0.9867 |
| GraphARM | 90.20 | - | - | 1.220 | - | - | - |
| GEEL | **100.0** | 96.08 | 22.30 | 0.089 | 0.9386 | 0.5161 | 0.9891 |
| *Molecule-specific generative models* | | | | | | | |
| CharRNN | 99.57 | - | - | **0.087** | 0.9313 | 0.5162 | 0.9887 |
| CG-VAE | **100.0** | - | - | 1.852 | 0.6628 | 0.3940 | 0.9484 |
| MoFlow | 91.36 | 98.65 | 94.72 | 4.467 | 0.1447 | 0.3152 | 0.6991 |
| **STGG** | **100.0** | 96.76 | 72.73 | 0.585 | 0.9416 | **0.9998** | **0.9984** |
| *Unconditional (masking all properties)* | | | | | | | |
| **STGG+** | **100.0** | 97.17 | 74.41 | 0.089 | 0.9265 | 0.5179 | 0.9877 |
| *Conditional (using test properties)* | | | | | | | |
| **STGG+** (k=1) | **100.0** | 97.63 | 75.99 | 0.134 | 0.8906 | 0.5004 | 0.9799 |
| **STGG+** (k=5) | **100.0** | 96.86 | 74.18 | 0.166 | 0.9050 | 0.5039 | 0.9860 |

Table 8: Molecular graph generation performance on Zinc250K.

| Method | Valid (%) (↑) | Unique (%) (↑) | Novel (%) (↑) | FCD (↓) | Scaf. (↑) | SNN (↑) | Frag. (↑) |
|---|---|---|---|---|---|---|---|
| *Domain-agnostic graph generative models* | | | | | | | |
| EDP-GNN | 63.11 | 99.79 | **100.00** | 16.737 | 0.0000 | 0.0815 | 0.0000 |
| GraphAF | 68.47 | 98.64 | 99.99 | 16.023 | 0.0672 | 0.2422 | 0.5348 |
| GraphDF | 90.61 | 99.63 | **100.00** | 33.546 | 0.0000 | 0.1722 | 0.2049 |
| GDSS | 97.01 | 99.64 | **100.00** | 14.656 | 0.0467 | 0.2789 | 0.8138 |
| DiGress | 94.99 | 99.97 | 99.99 | 3.482 | 0.4163 | 0.3457 | 0.9679 |
| DruM | 98.65 | 99.97 | 99.98 | 2.257 | 0.5299 | 0.3650 | 0.9777 |
| GraphARM | 88.23 | - | - | 16.260 | - | - | - |
| GEEL | 99.31 | 99.97 | 99.89 | 0.401 | 0.5565 | 0.4473 | 0.9920 |
| *Molecule-specific generative models* | | | | | | | |
| CharRNN | 96.95 | - | - | 0.474 | 0.4024 | 0.3965 | **0.9988** |
| CG-VAE | **100.0** | - | - | 11.335 | 0.2411 | 0.2656 | 0.8118 |
| MoFlow | 63.11 | 99.99 | **100.00** | 20.931 | 0.0133 | 0.2352 | 0.7508 |
| **STGG** | **100.0** | 99.99 | 99.89 | **0.278** | **0.7192** | **0.4664** | 0.9932 |
| *Unconditional (masking all properties)* | | | | | | | |
| **STGG+** | **100.0** | 99.99 | 99.94 | 0.395 | 0.5657 | 0.4316 | 0.9925 |
| *Conditional (using test properties)* | | | | | | | |
| **STGG+** (k=1) | **100.0** | **100.0** | 99.98 | 0.514 | 0.5302 | 0.4099 | 0.9917 |
| **STGG+** (k=5) | **100.0** | **100.0** | **100.0** | 0.562 | 0.5491 | 0.4176 | 0.9909 |

## A.8 ABLATION (ZINC)

Table 9: Ablation of MinMAE for out-of-distribution ($\mu \pm 4\sigma$) property-conditional generation on Zinc.

| | logP | | QED | | molWt | |
|---|---|---|---|---|---|---|
| Condition | 84.000 | 580.00 | -3.2810 | 8.1940 | 0.1778 | 1.2861* |
| Base | 0.0575 | 0.0751 | 0.0079 | 0.1946 | 0.0153 | 0.0008 |
| standardized properties | 0.0034 | 0.0943 | 0.0015 | 0.0072 | 0.0001 | 0.0004 |
| Randomize-order | 0.0034 | 0.0651 | 0.0001 | 0.0001 | 0.0002 | 0.0002 |
| Architecture | 0.0034 | 0.0086 | 0.0006 | 0.0001 | 0.00001 | 0.4042 |
| MLP (instead of a single-layer) | 0.0034 | 0.0183 | 0.0001 | 0.0007 | 0.0002 | 0.2277 |
| Property-prediction loss | 0.0034 | 0.0086 | 0.0004 | 0.0024 | 0.00005 | 0.3645 |
| Random-guidance | 0.0211 | 0.0240 | 0.0001 | 0.0007 | 0.000007 | 0.0058 |
| Filtering ($k = 5$) | 0.0010 | 0.0061 | 0.0000002 | 0.0281 | 0.0001 | 0.0012 |

*The value of 1.2861 is improper; we calculate the MAE with respect to the maximum QED (0.948).

## A.9 FULL TABLE OF CONDITIONAL GENERATION ON HIV, BBBP, AND BACE

Table 10: Full table: Conditional generation of 10K molecular compounds on HIV, BBBP, and BACE.

| Tasks | Model | Validity ↑ | Distribution Learning | | | | Condition Control | |
|---|---|---|---|---|---|---|---|---|
| | | | Coverage* ↑ | Diversity ↑ | Similarity ↑ | Distance ↓ | Synthe. MAE ↓ | Property Acc.* ↑ |
| Synth. & BACE | DiGress | 0.351 | 8/8 | 0.886 | 0.694 | 24.656 | 2.068 | 0.506 |
| | DiGress v2 | 0.355 | 8/8 | 0.881 | 0.703 | 25.327 | 2.337 | 0.511 |
| | GDSS | 0.288 | 4/8 | 0.876 | 0.271 | 46.754 | 1.642 | 0.504 |
| | MOOD | 0.995 | 8/8 | 0.890 | 0.259 | 44.239 | 1.885 | 0.506 |
| | Graph DiT | 0.867 | 8/8 | 0.824 | 0.875 | 7.046 | 0.400 | 0.913 |
| | Graph GA | 1.000 | 8/8 | 0.859 | 0.981 | 7.410 | 0.963 | 0.469 |
| | MARS | 1.000 | 8/8 | 0.834 | 0.883 | 6.792 | 1.012 | 0.518 |
| | LSTM-HC | 0.997 | 8/8 | 0.815 | 0.798 | 17.559 | 0.921 | 0.582 |
| | JTVAE-BO | 1.000 | 6/8 | 0.668 | 0.728 | 30.470 | 0.992 | 0.463 |
| | STGG** | 1.000 | 8/8 | 0.824 | 0.979 | 3.824 | 0.453 | 0.949 |
| | **STGG+** ($k = 1$) | 1.000 | 8/8 | 0.829 | 0.979 | 3.796 | 0.238 | 0.912 |
| | **STGG+** ($k = 5$) | 1.000 | 8/8 | 0.826 | 0.979 | 3.802 | 0.178 | 0.926 |
| | **Train data** | 1.000 | 8/8 | 0.819 | 0.981 | 3.837 | 0.003† | 0.991 |
| | **Test data** | 1.000 | **7/8*** | 0.824 | 1.000 | 0.000 | 0.002† | **0.817*** |
| Synth. & BBBP | DiGress | 0.696 | 9/10 | 0.910 | 0.681 | 18.692 | 2.366 | 0.654 |
| | DiGress v2 | 0.689 | 9/10 | 0.911 | 0.634 | 19.450 | 2.269 | 0.653 |
| | GDSS | 0.622 | 3/10 | 0.842 | 0.267 | 39.944 | 1.379 | 0.504 |
| | MOOD | 0.801 | 9/10 | 0.927 | 0.172 | 34.251 | 2.028 | 0.490 |
| | Graph DiT | 0.847 | 9/10 | 0.886 | 0.933 | 11.851 | 0.355 | 0.942 |
| | Graph GA | 1.000 | 9/10 | 0.895 | 0.951 | 10.166 | 1.208 | 0.302 |
| | MARS | 1.000 | 8/10 | 0.864 | 0.770 | 10.979 | 1.225 | 0.519 |
| | LSTM-HC | 0.999 | 8/10 | 0.888 | 0.893 | 16.390 | 0.997 | 0.559 |
| | JTVAE-BO | 1.000 | 5/10 | 0.746 | 0.582 | 33.575 | 1.162 | 0.496 |
| | STGG** | 1.000 | 9/10 | 0.891 | 0.916 | 11.736 | 0.982 | 0.754 |
| | **STGG+** ($k = 1$) | 1.000 | 10/10 | 0.888 | 0.937 | 9.859 | 0.466 | 0.867 |
| | **STGG+** ($k = 5$) | 1.000 | 9/10 | 0.887 | 0.936 | 10.101 | 0.381 | 0.900 |
| | **Train data** | 1.000 | 8/10 | 0.883 | 0.957 | 9.890 | 0.017† | 0.996 |
| | **Test data** | 1.000 | **10/10*** | 0.880 | 0.998 | 0.000 | 0.018† | **0.806*** |
| Synth. & HIV | DiGress | 0.438 | 22/29 | 0.919 | 0.856 | 13.041 | 1.922 | 0.534 |
| | DiGress v2 | 0.505 | 24/29 | 0.919 | 0.848 | 13.400 | 1.593 | 0.533 |
| | GDSS | 0.693 | 4/29 | 0.782 | 0.103 | 45.342 | 1.252 | 0.483 |
| | MOOD | 0.288 | 29/29 | 0.928 | 0.136 | 32.352 | 2.314 | 0.511 |
| | Graph DiT | 0.766 | 28/29 | 0.897 | 0.958 | 6.022 | 0.309 | 0.978 |
| | Graph GA | 1.000 | 28/29 | 0.899 | 0.966 | 4.442 | 0.984 | 0.604 |
| | MARS | 1.000 | 26/29 | 0.876 | 0.652 | 7.289 | 0.969 | 0.646 |
| | LSTM-HC | 0.999 | 13/29 | 0.909 | 0.915 | 7.466 | 0.948 | 0.674 |
| | JTVAE-BO | 1.000 | 3/29 | 0.806 | 0.417 | 41.977 | 1.236 | 0.485 |
| | STGG** | 1.000 | 27/10 | 0.899 | 0.961 | 4.558 | 0.442 | 0.950 |
| | **STGG+** ($k = 1$) | 1.000 | 27/29 | 0.896 | 0.970 | 4.075 | 0.314 | 0.876 |
| | **STGG+** ($k = 5$) | 1.000 | 24/29 | 0.897 | 0.9700 | 4.317 | 0.229 | 0.905 |
| | **Train data** | 1.000 | 27/29 | 0.895 | 0.970 | 4.019 | 0.018† | 0.999 |
| | **Test data** | 1.000 | **21/29*** | 0.895 | 0.998 | 0.074 | 0.015† | **0.726*** |

*The classifier from Liu et al. (2024) (used in the last column) has limited accuracy on the test set; thus, any *Property Acc.* above the **test data accuracy** is not indicative of better quality. Similarly, atom coverage is not 100% on test data; thus, any coverage above the **test set coverage** does not indicate better performance.
**STGG with categorical embedding, missing indicators, random masking, and extra symbol for compounds.
†The dataset properties are rounded to two decimals hence MAE is not exactly zero.

## A.10 FULL TABLE OF REWARD MAXIMIZATION ON QM9

Table 11: Reward maximization on QM9.

|  | Type | Data | Reward (↑) | Diversity (↑) |
|---|---|---|---|---|
| Envelope QL | | | 0.65 | 0.85 |
| MOReinforce | Online | 1M molecules | 0.57 | 0.53 |
| MOA2C | | | 0.61 | 0.39 |
| MOGFN-PC | | | 0.76 | 0.93 |
| STGG** | | | 0.73 | 0.10 |
| **STGG+** $(k=1)$ | Offline | QM9 ($\sim$115K molecules) | 0.78 | 0.76 |
| **STGG+** $(k=5)$ | | | 0.78 | 0.90 |
| **STGG+** $(k=100)$ | | | 0.77 | 0.98 |

**STGG with missing indicators, and random masking.

## A.11 BEST MOLECULES GENERATED BY STGG

### A.11.1 ZINC OOD

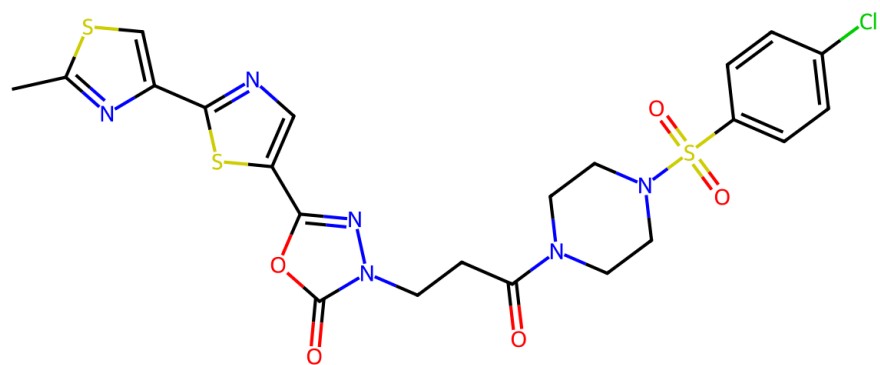

Figure 3: Conditioning on molWt=580.00

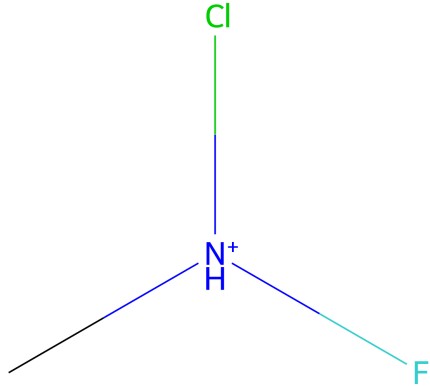

Figure 4: Conditioning on molWt=84.0008

Figure 5: Conditioning on logP=8.1940

Figure 6: Conditioning on logP=-3.2810

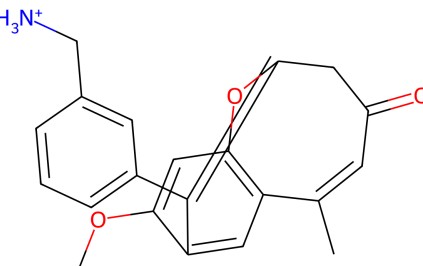

Figure 7: Conditioning on QED=1.2861

Figure 8: Conditioning on QED=0.1778 (which means low drug-likeness and less "chemical beauty" (Bickerton et al., 2012))

### A.11.2   QM9 REWARD MAXIMIZATION

Figure 9: Best QM9 reward maximization molecules

### A.11.3 CHROMOPHORE OOD

Figure 10: Conditioning on molWt=1538.00

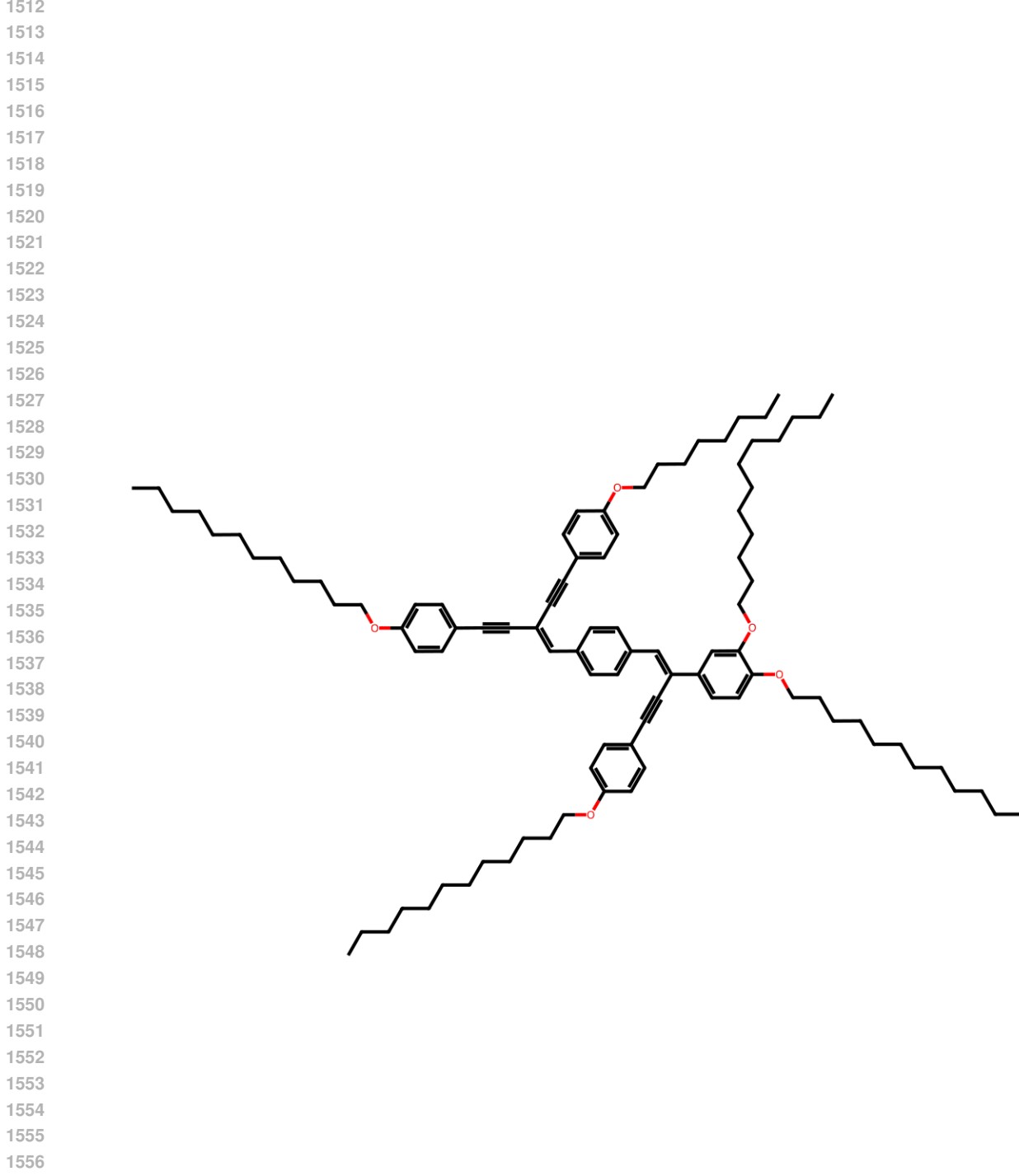

Figure 11: Conditioning on logP=28.6915

Figure 12: Conditioning on logP=-13.6292

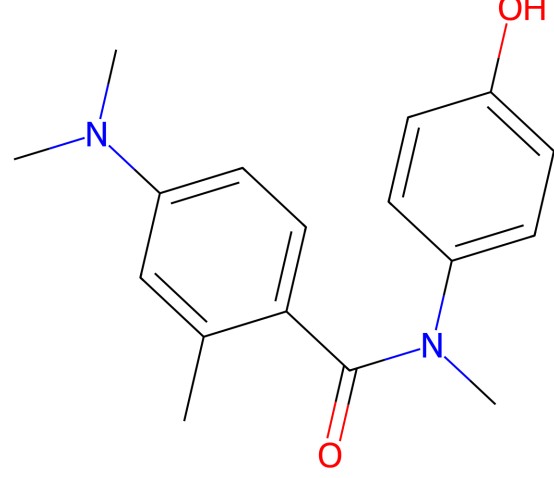

Figure 13: Conditioning on QED=1.2355

## A.12 OOD Tables with top-100 MAE instead of MinMAE

In this section, we report the average of the top-100 molecules MAE instead of the top-1 MAE (MinMAE) for the out-of-distribution (OOD) Tables. We report the MinMAE in the main manuscript since this is the metric reported by Kwon et al. (2023) baselines. Since Kwon et al. (2023) did not release their code or test on top-100 MAE, the VAE baselines are not included in the tables below.

In Table 12, STGG+ performs much better than STGG in all cases except for high QED, where STGG is slightly better. Random guidance is helpful for high QED and logP.

Table 12: Out-of-distribution ($\mu \pm 4\sigma$) property-conditional generation of 2K molecules on Zinc250K. Top-100 MAE.

| | Properties - top-100 MAE | | | | | |
| | molWt | | logP | | QED | |
| Condition | 84 | 580 | -3.2810 | 8.1940 | 0.1778 | 1.2861* |
| --- | --- | --- | --- | --- | --- | --- |
| STGG** | 18.248 | 5.559 | 1.204 | 1.548 | 0.206 | 0.022 |
| **STGG+** ($k=1$) | 0.790 | 1.389 | 0.018 | 0.900 | 0.003 | 0.561 |
| **STGG+** ($k=5$) | 1.289 | 1.503 | 0.021 | 3.710 | 0.003 | 0.571 |
| **STGG+** ($w \sim \mathcal{U}(-0.5, 2), k=1$) | 1.533 | 2.088 | 0.040 | 0.285 | 0.005 | 0.060 |
| **STGG+** ($w \sim \mathcal{U}(-0.5, 2), k=5$) | 1.285 | 1.104 | 0.022 | 0.803 | 0.004 | 0.042 |

*The value is improper; we condition on 1.2861 but calculate the MAE with respect to the maximum QED (0.948).
**STGG with missing indicators, and random masking.

In Table 13, STGG+ with pre-training and fine-tuning generally performs slightly better than regular training. Random guidance is helpful for high QED.

Table 13: Out-of-distribution ($\mu \pm 4\sigma$) property-conditional generation of 100 molecules on Chromophore DB. Top-100 MAE. We removed the low molWt and QED which are both impossible negative values.

| | Properties - top-100 MAE | | | |
| | molWt | logP | | QED |
| Condition | 1538.00 | -13.63 | 28.69 | 1.24* |
| --- | --- | --- | --- | --- |
| Trained on Chromophore DB (1000 epochs) | | | | |
| **STGG+** ($k=1$) | 256.6 | 11.1 | 5.1 | 0.6 |
| **STGG+** ($k=100$) | 562.3 | 11.0 | 16.3 | 0.5 |
| **STGG+** ($w \sim \mathcal{U}(-0.5, 2), k=1$) | 805.6 | 15.4 | 11.1 | 0.5 |
| **STGG+** ($w \sim \mathcal{U}(-0.5, 2), k=100$) | 609.5 | 8.8 | 14.3 | 0.2 |
| Pre-trained on Zinc250K (50 epochs) and fine-tuned on Chromophore DB (100 epochs) | | | | |
| **STGG+** ($k=1$) | 294.9 | 8.4 | 6.1 | 0.5 |
| **STGG+** ($k=100$) | 401.9 | 5.6 | 13.1 | 0.4 |
| **STGG+** ($w \sim \mathcal{U}(-0.5, 2), k=1$) | 543.0 | 14.6 | 12.7 | 0.5 |
| **STGG+** ($w \sim \mathcal{U}(-0.5, 2), k=100$) | 416.5 | 6.1 | 13.0 | 0.2 |

*The value of 1.24 is improper; we calculate the MAE with respect to the maximum QED (0.948).

