# OpenReview forum: "Any-Property-Conditional Molecule Generation with Self-Criticism using Spanning Trees"
_ICLR.cc/2025/Conference — Submitted to ICLR 2025_

### Official Review · Reviewer_atbB · 2024-11-01

**Soundness:** 3
**Presentation:** 4
**Contribution:** 2
**Rating:** 5
**Confidence:** 4

**Summary:**

This paper tackles the issue of generating valid molecules by extending the Spanning Tree-based Graph Generation (STGG) to support multi-property conditional generation. The proposed STGG+ integrates a Transformer architecture, property masking, and an auxiliary loss for model self-evaluation.

**Strengths:**

1. The paper is well written.
2. The authors suggest modifications that make sense and feel intuitive.

**Weaknesses:**

1. I find the changes to be intuitive but somewhat obvious, leading me to believe that the paper lacks significant novelty.
2. Please see questions.

**Questions:**

1. I didn't completely follow the details of best-of-k filtering described in Section 3.6. It would be good if the authors could explain how this is exactly done.
2. Based on observations: In Table 1, STGG+ shows an improved synth MAE, but other metrics appear comparable. In Table 2, STGG+ outperforms by design. In Table 3, the settings aren't directly comparable. Does this imply that the primary novelty lies in achieving property conditioning? If so, is the novelty somewhat limited, given that the extension feels intuitive?
3. Considering the non-comparable nature of Table 3, could the experiments be repeated under consistent settings for a fair comparison?
4. It’s currently unclear which modifications specifically drive the improvements in STGG+. Conducting ablation studies based on the points listed on page 2 (points 1-5) would provide more clarity.
5. For conditional generation, online methods like GFlowNets could serve as an additional baseline. Would it be possible to include this baseline in Table 2?

**Paper suggestions:**

a) I think, including Figure 3 (suplementary) instead of Figure 1 in the main paper would better showcase the contributions.

b) For section 3.6, I found the figure a little confusing. It may help to have a better figure to explain the same.

---

> ### Author Response · Authors · 2024-11-15
> **atbB**
>
> Thank you for your review. We address your questions and comments below.
>
> > Weaknesses
>
> While our approach is intuitive, it also introduces several novel improvements. We propose many improvements, such as guidance in autoregressive models (extremely rarely used in the autoregressive literature), random guidance (we are unaware of prior works using this technique), self-property-predictor (we are not aware of molecule generation models using this, but there are likely similar ideas in the LLM world).
>
> > Questions
>
> 1) We updated Figure 2 to make self-criticism more clear (per your suggestion). Explanation: During training we predict the properties of the molecule. At generation we generate K molecules conditional on the desired properties. Then we mask the properties, effectively removing this information, and the model predict the properties. Out of the K molecules, we keep the best one, i.e., the molecule whose predicted properties are closest to the desired properties.
>
> 2) To clarify, we propose many novelties (as mentioned above) and we use them in order to improve property conditioning performance. Table 1 (especially the full version in Table 9, Appendix A.9) shows that our model generally has the best Validity, coverage, Frechet distance and property accuracy as well. Can you elaborate on why you believe that in Table 2, STGG+ outperforms by design?
>
> 3) They are inherently different approaches to solve a similar problem. We agree that Table 3 comparison is thus somewhat of an apples-to-oranges comparison.  Even if the settings are different, we performed this comparison to see if generative models could perform well, as they have the advantage of being able to generate molecules for different property combinations without needing to retrain with a different reward. We believe that having this comparison, even if imperfect, is better than not.
>
> 4) We have ablations in Table 8 of Appendix A.8.
>
> 5) In theory it would be possible to train GFlowNet models on the 6 different OOD properties for Table 2. However, this would require a property predictor (increasing the complexity of this comparison), because if GFlowNet were to use RDKit directly to measure the true property, then it is an unfair advantage in terms of the effective dataset size. We only made the RL/GFlowNet comparison in Table 3 in order to have one small comparison with these methods. It would be more suitable to have a separate paper focusing on comparing RL/GFlowNet to generative models as this is not our main goal.
>
> > Paper suggestions
>
> a) We initially thought that Figure 3 (supplementary) was a bit too big and could confuse the reader, which is why we had put it in the Appendix. But since you suggested putting it in the main text, we gladly changed the main Figure to Figure 3 (supplementary).
>
> b) Per your suggestion, we made Figure 2 much simpler and more clear. This significantly improves readability.

---

> > ### Comment · Reviewer_atbB · 2024-11-26
> >
> > Thank you for your responses. I have gone over other reviews and responses as well and have currently decided to maintain my score for the time being. I am waiting for Reviewer WwA3's comments on your response, post which I will update my score accordingly.

---

> > > ### Comment · Reviewer_atbB · 2024-12-03
> > >
> > > I have followed Reviewer WwA3 observations and I agree with their sentiment. Hence I will keep my current score.

---

### Official Review · Reviewer_d7sw · 2024-11-02

**Soundness:** 3
**Presentation:** 4
**Contribution:** 3
**Rating:** 5
**Confidence:** 2

**Summary:**

This paper considers the problem of generating (hopefully) new representations of molecules. Existing techniques either work with a 1D string representation of molecules or a 2D graph representation. This work considers the former representation and builds on a method called Spanning Tree-based Graph Generation (STGG) that was specifically created to use generative AI models. The present work differs from existing work along the following two high level aspects:

1. Previous work would generate molecules without any restrictions. However, this paper considers the problem of generating molecules that has to satisfy (some) subset of a given set of properties that the generated molecules must satisfy.
2. Unlike existing results the paper creates models that can _self-criticize_ by allowing the model to predict properties of the molecules it generates and uses that to prune out molecules that do not satisfy the required properties.

The paper lists conceptual improvements made in this work in the context of generating molecules but since I'm more familiar with Transformers and related literature, I will focus my review on those. Specific to the Transformer model used in this work (as opposed to the STGG work), the paper uses improvements made to the Transformer architecture (e.g. FlashAttention) over the last three years.

The paper presents a pretty comprehensive (at least to me) set of experiments and show that the proposed new system works better than existing systems on benchmarks that are used in this area.

**Strengths:**

* The problem considered in this paper (molecule generation) has clear practical importance and given that there exists standard 1D string representation of molecules, using generative AI to create new molecules is definitely a  promising avenue worth pursuing.

* The paper has a pretty comprehensive experimental results and they show the efficacy of the proposed system.

* While some of the techniques used in the paper might be `standard' for say language modeling, being able to apply these techniques in a completely new application domain and show improvement is very nice.

* The ability to impose certain properties on molecules being generated and having the model be able to self-criticize seems like really nice capabilities.

**Weaknesses:**

Below are some questions that I was not able to answer based on what is there in the paper. (Again, as mentioned earlier, I focused in the Transformer aspects of the paper so all of the questions below are on that axis.) Also some of these questions might be asking for intuitive answers and not necessarily something that could potentially be answered with experiments-- but having these answers might be useful for the reader to understand some of the design choices made by the paper:

* (Q1) The paper uses causal Transformer models-- is there any reason a non-causal Transformer model cannot be used? E.g. non-causal Transformer models have been used on applications other than language modeling (e.g. ViT in image processing)-- and pre-Transformer language models like BERT were non-causal. In theory non-causal models are more expressive than non-causal model (just because a causal model is trivially a non-causal model as well).

* (Q2) The idea of having the model self-criticize the molecules it generates reminded me a lot of GANs. Have GANs been tried to generate molecules? If so, have they worked or is there some intuition for why they did not work?

* (Q3) Over the last ~3 years there have been a fair amount of work on `Transformer-free' models for language generation. One such line of work is based on state space model (Mamba [see https://arxiv.org/abs/2405.21060 and https://arxiv.org/abs/2312.00752] being a model that has garnered a lot of attention in the language modeling literature). Some of these ideas have been used in genomic sequencing (e.g. Hyena DNA-- https://arxiv.org/abs/2306.15794). Where these recent models considered in this work?

* (Q4) In lines 227-228, it is mentioned that the _number_ of masked properties $t$ was picked uniformly at random between $0$ and $T$. However, which of the $\binom{T}{t}$ subset of properties were actually chosen to mask?

Below are some minor comments (that are purely related to presentation):

* Lines 354-355: Instead of saying "similar" performance-- please quantify, i.e. within what percentage of existing work?

* Table 1: Is _Distance_ in the table column name the same as FCD?

**Questions:**

Please address (to the extent possible) Q1-Q4 in the Weakness section. Specifically for Q1-Q3, please state if the alternate techniques were considered when designing the experiments/writing the paper. If so, please explain why those alternate ideas were not incorporated into the paper. If not, please discuss how these alternative techniques might be relevant to the problem considered in the paper.

Post-Rebuttal Comments
----------------------------

The authors have addressed pretty much all of my concerns. As of Nov 25, I'm keeping my original score since I'm curious to see how the reviewers with the two lowest scores (one of whom is an expert in the paper's area) respond to the author's responses to their questions.

---

> ### Author Response · Authors · 2024-11-15
> **d7sw**
>
> Thank you for your review. We address your questions and comments below.
>
> Q1) STGG originally used a causal Transformer. We could use non-causal Transformer but for this we would need to switch to diffusion or use BERT-like masking. But in this case, we generate one atom and vertex after another in a random order, so we must follow causality in order to autoregressively generate the molecule. A limitation of diffusion is that we would not be able to use the powerful masking from STGG which masks future invalid tokens before the softmax. STGG masking is very powerful, so we pushed into this direction.
>
> Q2) We were not very familiar with GANs for molecule generation, so we couldn’t say for sure. But in practice, discrete data (text, molecules, etc.) tends to work better with autoregressive models than with GANs. Hence why GANs are very rarely used for chatbots anymore.
>
> Q3) We did initially consider Mamba, RKWV, and similar recurrent/SSSM models to improve inference speed. However, after consulting with chemists, we learned that it is extremely hard to synthesize molecules of very large sizes (>120 atoms for example), so the context length is limited (e.g., the largest molecule had 511 tokens on ChromophoreDB compared to the >=4096 context-length we normally see in LLMs). As long as context length is smaller than let say 1024-2048, FlashAttention is fast enough that there is no inference speed benefit for using these recurrent/SSSMs models.
>
> Q4) Each time we sample a training molecule, we choose a random number t of properties to mask uniformly between 0 and T, then we randomize the order of the properties and mask the first t properties (thus, a random subset is chosen).The code looks like this:
>         	batch_choices = torch.arange(n_properties).unsqueeze(0).repeat(batch_size,1) < torch.randint(0, n_properties+1, (batch_size,1)) # random choose how many properties to keep (equal-prob of each amount of properties)
>         	batch_choices = shufflerow(batch_choices, 1) # shuffle [b, n_properties] to randomize which properties are selected
>
> We addressed your comments: 1) We added a shortened table comparing all 3 best methods for unconditional generalization to be more quantitative in how we compare them; we still left the full table in the Appendix. This is more clear and easy for the reader to follow. 2) We changed the header to FCD to clarify that Distance stands for FCD.
>
> >Question:
>
> We give a brief explanation of the path that led to this direction in the Background section. To be more specific: We chose STGG after doing an extensive literature review of molecules generation papers. We found that many papers  recently focused on diffusion models respecting equivariance. However, the older SMILES-based models were performing just as well, which was surprising to us given all this time spent by researchers on newer diffusion methods. Out of the SMILES autoregressive methods, we found a slight variant: STGG, which is effectively a slightly modified SMILES vocabulary with the powerful masking of invalid tokens which improves performance. We found STGG to perform the best or among the best among many molecule generation datasets (see “Jang, Yunhui, Seul Lee, and Sungsoo Ahn. "A Simple and Scalable Representation for Graph Generation." The Twelfth International Conference on Learning Representations. 2023.” and “Ahn, Sungsoo, et al. "Spanning tree-based graph generation for molecules." International Conference on Learning Representations. 2021.
> “). This was the clear winner in our point of view. So we seeked to extend it, solve some of its remaining limitations, and improve it so that we could push it to a practical level. We also found STGG+ can perform very well in real-world applications (on proprietary datasets).

---

> > ### Comment · Reviewer_d7sw · 2024-11-17
> > **Thanks!**
> >
> > Thanks for your responses. Please include the  responses to my Q3 and Q4 in your draft since I think those clarifications will be useful for the readers.
> >
> > I have yet to go through the other reviews (and y'all's responses) to them carefully. After reviewing those, I'll decide if I should change my rating.

---

> > > ### Author Response · Authors · 2024-11-18
> > > **Updated**
> > >
> > > Thank you for the quick response. We updated the paper to include responses from Q3 and Q4. See line 204 and line 216 both pointing to the new details added to the Appendix.

---

### Official Review · Reviewer_cWrJ · 2024-11-02

**Soundness:** 3
**Presentation:** 2
**Contribution:** 3
**Rating:** 3
**Confidence:** 3

**Summary:**

This paper proposes an enhanced version of Spanning Tree-based Graph Generation (STGG+), tailored for multi-property conditional molecule generation. Based on the STGG, STGG+ includes improvements in the Transformer architecture, a flexible conditioning mechanism for any subset of properties, and a self-criticism mechanism that filters generated molecules based on a property predictor.

**Strengths:**

This paper used a property-predictor-driven self-criticism mechanism that allows STGG+ to evaluate and select the best out of multiple generated molecules, improving fidelity to target properties.

**Weaknesses:**

1. The model’s effectiveness relies heavily on the internal property predictor, which may be less reliable for out-of-distribution samples. This dependence could reduce fidelity in less representative scenarios.
2. Although the model improves conditioning performance, it’s unclear how it balances molecule diversity and property, diversity is also a crucial metric in molecular generation.

**Questions:**

1. The authors improved the structure of the original Transformer, but the results do not seem to reflect the improvements, such as whether the generation time and the quality of the generated molecules have been improved.
2. For the self-criticise, the authors should discuss the trade-off between performance gains and computational cost. Including a comparison of computational time for different values of k would clarify the model's efficiency.
3. The authors should optimize the structure of the result table, as it is not clear what is being compared, e.g. modify the table head.
4. For property-conditional generation. The authors only compare the MinMSE and should add some property distributions to demonstrate that the generated molecules approximate the given conditions.

---

> ### Author Response · Authors · 2024-11-15
> **cWrJ**
>
> Thank you for your review. We address your questions and comments below.
>
> > Weaknesses
>
> 1) The method still performs very well without the self-property-predictor (see k=1 in the Tables). In general, we found the self-property-predictor to perform well on most OOD properties, but in some cases, less so. As mentioned in lines 440-443 and 534-535, we found performance good except for OOD high logP conditioning values.
>
> 2) Just to be clear, we evaluated coverage, diversity, and similarity to training molecules in Table 1. We evaluated the % of valid, novel, and unique molecules (efficiency) in Table 2 and 4 (novelty means that the generated molecules are not found in the training dataset). We also look at Tanimoto diversity in Table 3. In all of these Tables, we see that our model performs well both in diversity metrics and property at the same time (efficiency close to 1 on Table 2 and Table 3, and highest coverage with high diversity in Table 1).
> It is possible to balance diversity and property fidelity trade-offs by changing the guidance and temperature (lower temperature (e.g., 0.7) leads to higher diversity, and lower guidance (e.g., 0.8) pushes less towards the property conditioning).
>
> > Questions:
> 1) In all experiments we compare STGG and STGG+ showing improvements. We also have ablations in Table 8 (Appendix A.8) starting from STGG and incrementally adding features to make it STGG+. The generation time is generally unimportant as it is sufficiently fast; there is no significant speed difference in our experience.
>
> 2) The compute cost is directly proportional to k, so k=5 makes the generation 5 times slower. But ultimately generation time is not a big factor since we are dealing with objects of at most 511 tokens (the largest molecule in Chromophore Db). We are very far from the LLM world which deals with > 4096 tokens and large models (we use only 3 layers!).
>
> 3) Can you point us to the table's column heads that are unclear? Another reviewer asked for clarifications to Table 1 header, so we will update these ones.
>
> 4) To clarify, Table 1 is the MAE (instead of the MinMAE) so it compares all generated molecules properties to the conditioning property. Also the Distance in Table 1 is the Frechet Distance which measures a distributional distance between training and generated molecules. For Table 3 we agree that it would be preferable to also show the MAE over top-10 or top-100 molecules in addition to the MinMAE. However, we reached out to the author of the VAE baselines and they didn’t have access to the code anymore. Replicating is non-trivial, so we focused on the top-1 which is the only metric used in the paper with the VAE baselines.

---

> > ### Author Response · Authors · 2024-12-03
> > **Response 2**
> >
> > Dear reviewer cWrJ,
> >
> > The paper has been significantly improved through your feedback. Did our response adequately address your previous concerns?
> >
> > Also note that since the last response, we have improved the paper in many ways:
> > - changed Fig 1 to highlight our contributions over STGG
> > - mention that we randomize the order of the molecules and that this increases generalization over STGG
> > - explain better the random guidance through balancing exploration and exploitation
> > - simplified Figure 2 for better understanding
> > - clarified which property predictor is used for each property (RDKIT unless otherwise specified)
> > - added the unconditional generation table (Table 1) to the main paper
> > - cleaned Table 2 headings and details to make clear what are the metrics and how they work
> > - added a row to Table 3 and 5 showing the "Training data (closest real sample)" for the MinMAE. This shows that no training sample is close to the +-4SD desired properties, thus our model extrapolates well beyond what is known.
> > - added information in the Appendix A.3 on canonicalization (how STGG molecules are converted back to canonical SMILES)
> > - added details on how the property masking works in the Appendix A.3
> > - discuss other potential architectures that we also considered in the Appendix A.5
> > - added the top-100 MAE results for Table 3 and 5 in Appendix A.12
> > - revised a few typos
> >
> > Thank you for your time.

---

### Official Review · Reviewer_WwA3 · 2024-11-03

**Soundness:** 2
**Presentation:** 2
**Contribution:** 1
**Rating:** 3
**Confidence:** 4

**Summary:**

This paper proposed the STGG+ method, an improved method of the Spanning Tree-based Graph Generation (STGG), for generating novel molecules with multi-property conditional generation. Architecture-wise, this work introduced

1. an improved Transfomer with Flash-Attention. RMDProp, etc.
1. an extended STGG model for more robust graph generation of molecules.

By randomly masking some properties at training time and using Classifier-Free Guidance (CFG), the model was shown to generate novel in- and out-of-distribution molecules with any property conditional generation.

**Strengths:**

### Originality
1. This work proposed an improved method of the STGG model for any property conditional molecule generation.
1. This work introduced classifier-free guidance and self-criticism into the transformer architecture.

### Quality
1. The proposed method is shown to improve the validity and diversity of generated molecules.
1. The method is also shown to better generate molecules with desired/conditioned properties.
1. The results are shown across multiple datasets and properties.

### Clarity
1. The STGG+ architecture is clearly explained.

### Significance
1. Any-property conditional generation is a challenging yet important task in technical applications. For instance, in drug discovery, it is important to generate molecules with desired curative properties and avoid molecules with toxic properties.

**Weaknesses:**

As a computational chemist with expertise in molecules (and SMILES), I am concerned with
1. the contribution and improvement of this work, STGG+, to the original STGG model or the original SMILES-based molecule generation methods.
2. this work's representation of chemistry and molecules in terms of correctness and novelty.

**The STGG+ representation of molecules is within the capabilities of SMILES representation.**
1. The STGG+ improvements to STGG are not significant enough
   - the proposed improvements such as masking of invalid tokens, automatic calculation of valency, etc., seem similar to adding if-else conditions to improve the original STGG model.
   - In my opinion, these improvements should be learned by the model itself during training. The model itself should learn to avoid invalid tokens and keep track of valency. These are all fundamental grammar rules that the model should learn by itself.
   - If these constraints are manually added, it limits the efficiency of the generation process. For example, valency calculation can be intricate in the generation process when rings are involved.
   - Because of these manual implementations, I am not convinced that the STGG+ representation is significantly better than the SMILES representation in terms of ensuring valid molecules.
2. The proposed benefits of STGG+ (spanning-tree representation) compared to SMILES representation are not entirely true. SMILES can also achieve the claimed benefits with similar modifications. For example,
   - In SMILES, rings are represented by two identical numbers at the beginning and end of the ring. For cyclohexene, its SMILES representation can be `C1CCCCC=1` (or `C1-C-C-C-C-C=1`) and its spanning-tree representation can be `[bos]C[bor]-C-C-C-C-C=[eor1][eos]`. In the spanning-tree representation, a `[bor]` token must be paired with a `[eor#]` token before `[eos]` to form a valid molecule. Whereas in SMILES, a ring-starting number must be paired with the same number before the end of the string.
   - Automatic calculation of valency can be done in SMILES as well in the same fashion as STGG+ since the spanning-tree representation and the SMILES representation are interchangeable during the generation process.

**This work lacks clarity on the spanning-tree representation such as explicit/implicit hydrogen atoms and canonical representation.**
1. In Figure 1, line 140, the spanning-tree representation used a combination of explicit and implicit hydrogen atoms. The nitrogen atom was shown with an explicit hydrogen atom, and all the carbon atoms were shown without hydrogen atoms (implicit). However, from Appendix A.4, it seems that the vocabulary is collected for spanning tree representations with explicit hydrogen atoms.
1. The above point leads to the question of canonical representation - The same molecule can have different SMILES representations and different spanning-tree representations. In other words, different sequences of tokens can point to the same molecule. For example, `[bos]C[bor]-C-C-C-C-C=[eor1][eos]`, `[bos]C[bor]-C-CH2-C-C-C=[eor1][eos]`, and `[bos]C[bor]-CH2-C-CH2-C-C=[eor1][eos]` can all represent the same cyclohexene molecule.
1. For the reported generative efficiency (% of valid, novel, and unique molecules), was canonicalization performed/considered? If not, the reported efficiency might be overestimated. Additionally, the authors should provide some examples of the generated sequences and their corresponding molecules to clarify the canonicalization process.
2. **Suggestions:** In Section 3.3, the authors should discuss the issue with explicit/implicit hydrogen atoms and canonical representation. Try to clarify the following points:
   - Does STGG+ generate molecules with explicit hydrogen atoms only or can it also generate molecules with implicit hydrogen atoms? The spanning-tree representation should allow both.
   - How is the canonical representation handled in the training and generation processes?
   - **What is the definition of valid, novel, and unique molecules?** Are molecules considered unique if they have different sequences of tokens but represent the same molecule?

**The reported property MAE of the conditional generation needs more explanation.**
1. For the properties of the generated molecules, were they calculated with the property predictor of the STGG+ architecture or with an external property calculator such as RDKit? The external predictor should provide the ground truth for the property values and should thus be used for the evaluation.
2. The `MinMAE` reported in Table 2 needs more clarification: is it the minimum absolute error across the 2K generated molecules? What does "minimum mean" refer to?
   - If the minimum is reported, what about the mean of the absolute errors?
   - For such a large number of generated molecules, the mean absolute error is a better metric to evaluate the performance of the conditional generation. This is related to the application of the model (line 57) - validating the properties of the generated molecules in real life can be costly. Conditional generation aims to generate a small set of potential candidates. Minimum error implies that one has to test all 2K molecules (too large) to find the best candidate, while average error better represents the overall conditional generation performance.
   - The minimum absolute error might be more convincing if reported on a small population of generated molecules such as 10x molecules with multiple batches.

**Questions:**

My questions are closely related to the weaknesses mentioned above. The authors are encouraged to address the points raised in the weaknesses section. Some questions include but are not limited to:
1. The STGG+ approach claims improvements over SMILES-based molecule generation methods. **What are some of the improvements that the SMILES representation fails to achieve or is difficult to achieve compared to the STGG+ representation?**
2. The spanning-tree representation allows both explicit and implicit hydrogen atoms. Are the generated molecules restricted to explicit hydrogen atoms or can they also have implicit hydrogen atoms?
3. How does the model/evaluation handle canonicalization of the generated sequences/molecules?
   - Does uniqueness consider canonicalization or does it only consider differences in the generated sequences?
4. For reporting the property MAEs, was an external property predictor used for the evaluation? How is MinMAE reported?
   - If an external property predictor was used, provide the details of the external predictor for MolWt, LogP, QED, and HOMO-LUMO gap.

---

> ### Author Response · Authors · 2024-11-15
> **WwA3**
>
> Thank you for your review. We address your questions and comments below.
>
> > The STGG+ representation of molecules is within the capabilities of SMILES representation
>
> We want to reiterate that the core contributions of our approach are  property-conditioning, improved architecture, improved spanning-tree,
>
> 1) We want to clarify that the “if-else conditions” is the main contribution of STGG over SMILES. It's through this masking with if-else conditions that STGG significantly improves the quality of generated molecules over SMILES. The representation is not necessarily better, but the STGG masking prevents invalid choices during inference rather than needing to discard invalid molecules after generation. The authors of STGG showed that it performs better than SMILES. We further extend and enhance STGG given its good performance. Your concerns are thus targeted specifically at STGG, which is the foundation we use. To reiterate, we made 5 sets of contributions (property conditioning, improved architecture, many improvements to STGG, auxiliary prediction loss for improved generalization and allowing us to do self-criticism, and classifier-free guidance with random sampling).
>
> 2) We agree that canonical SMILES could use the STGG masking, the STGG creator simply modified the SMILES slightly to make the masking a bit easier to define. Again, this is a concern  targeted specifically at STGG, not our work which just leverages and extends STGG. We use STGG because its performance was shown to be better than SMILES (See “Ahn, Sungsoo, et al. "Spanning tree-based graph generation for molecules." International Conference on Learning Representations. 2021.“).
>
> As a side note: The main novelty of STGG is the masking of tokens using a SMILES-like vocabulary for improved molecule validity. This is why we extend it to make it even better.
>
> > This work lacks clarity on the spanning-tree representation such as explicit/implicit hydrogen atoms and canonical representation.
>
> 1) Following STGG, the vocabulary uses explicit H’s, but after converting STGG to SMILES, we pass the SMILES to RDKit which can decide to ignore the explicit H and change the number of H. This is why the plots may be different since they are RDKit plots. We added this information in Appendix A.3.
>
> 2) There are many possible orderings in which to traverse the graph. Contrary to STGG, we do not canonicalize molecules, we use a random ordering of the molecules (a different random ordering is sampled for each molecule during training). Doing so improves generalization (see Table 8 in Appendix A.8). It is only mentioned on line 85 in ‘contributions’ that we use random ordering; we will mention it in the text because it should be made more clear.
>
> 3) We convert our STGG molecules to canonical SMILES after generation, thus novelty and uniqueness is correct. Thank you for the great suggestion; we added an example of the transition from SMILES to STGG to Canonical SMILES in Appendix A.3.
>
> 4) Training with only implicit H’s is expected to work, but we haven’t done this ourselves. We added these details to Appendix A.3.
>
> > The reported property MAE of the conditional generation needs more explanation.
>
> 1) We use RDKIT to evaluate the properties, except for Property Acc. in Table 1 which is based on a property-predictor as mentioned in the footnote of the table. We had also forgotten to mention that Table 3 uses MXMNet to evaluate the HOMO-LUMO Gap. We now mention all of this information more explicitly in the paper. We also revised the headers of Table 1 to make it more clear what the metrics are (as suggested by a reviewer).
>
> 2)
> - MinMAE means that for each generated molecule, we get the MAE between the generated molecule properties and the real properties (mean over all properties), then we report the minimum MAE (for the molecule closest to the true property). We prefer the MinMAE over the MeanMAE since our goal is to find new molecules with desired OOD properties. It is not important to our needs that the average molecule is close to the property, what matters most for material discovery is to find even just one such molecule that has the correct difficult-to-obtain set of properties. Finding the needle in the haystack is the ultimate goal.
> - On another note, we would like to point out that the VAE baselines have no open-source code; we reach out to the authors and they responded that they don’t have access to the code anymore, so replication would be extremely difficult, which is why we ended up only using the same metric that they used (MinMAE).
> - Note that Table 1, contrary to other tables, compares the pairwise generated molecules properties to real properties and average over all generated molecules so this specific table does not use the MinMAE and instead uses the MeanMAE.
>
> [1/2]

---

> > ### Author Response · Authors · 2024-11-15
> >
> > > Questions
> >
> > 1) STGG improves over SMILES through the masking to prevent invalid choices. We do not claim that STGG is a better representation than SMILES.
> >
> > 2) We chose explicit H only, no implicit. This is an arbitrary choice following STGG. We now mention it in Appendix A.3.
> >
> > 3) We always canonicalize to canonical SMILES after generation so uniqueness and novelty is properly calculated. As mentioned above, we added this to the Appendix.
> >
> > 4) We will add details about how the properties are calculated in the text. We currently use RDKIT unless otherwise specified.
> >
> > [2/2]

---

> > > ### Comment · Reviewer_WwA3 · 2024-11-26
> > >
> > > Thank the authors for the discussions.
> > >
> > > I am still not convinced of the idea that "STGG improves over SMILES through the masking to prevent invalid choices". The "STGG+" seems more of "SMILES+". In other words, in my opinion, similar improvements can be achieved with SMILES.
> > >
> > > Even if I agree that those are notable improvements, the results are not quite convincing either:
> > > 1. The chosen metrics such as molWt, logP, and QED are not exactly challenging. These can be empirically calculated and somewhat linearly related to the string representation itself - they are not challenging enough to learn or predict. HOMO-LUMO gap is arguably the most interesting property in the paper but not many results are shown.
> > > 2. MinMAE is a weak metric. The authors reported the minimum error based on 1000 generated molecules. To achieve the minimum error, **all 1000** molecules must be labeled and compared - this can be costly in real-life applications where labeling 1000 molecules can be consuming. In addition, in Table 3, the improvements are hardly convincing - the baseline MinMAE is already pretty low. Sure, the STGG+ decreased the error to a few magnitudes lower, but this seems more of a float-point accuracy improvement - the authors picked some values that can be exactly achieved by certain molecules and STGG+ found one.
> > >
> > > In a more basic case, what would the MinMAE be if the authors just randomly sampled 1000 molecules from the dataset? Of course, this basic case does not have OOD capability, but I believe a simple genetic algorithm (GA) can help. GA can even make sure that the generated molecules are 100% correct.
> > >
> > > The authors did not agree that MeanMAE should be more significant, but MeanMAE reflects the true capability of a model. MinMAE requires a global comparison amongst all generated molecules, but MeanMAE can reflect the performance of the model even if a smaller number of molecules is sampled.
> > >
> > > Due to my concerns, I am keeping my original score. I do not think that the paper added much novelty to the field.

---

> ### Author Response · Authors · 2024-11-27
> **Response**
>
> > I am still not convinced of the idea that "STGG improves over SMILES through the masking to prevent invalid choices". The "STGG+" seems more of "SMILES+". In other words, in my opinion, similar improvements can be achieved with SMILES.
>
> - We agree that STGG and SMILES are very similar, and that our improvements could have been based on a SMILES representation instead of STGG. We chose to improve STGG instead of SMILES because it is the state-of-the-art on many molecule generation datasets [1,2]. This does not change the fact that our proposed model shows improvements on a wide variety of tasks.
>
> > The chosen metrics such as molWt, logP, and QED are not exactly challenging - they are not challenging enough to learn or predict. HOMO-LUMO gap is arguably the most interesting property in the paper but not many results are shown.
>
> See our response to Reviewer GMmJ:
> - We fully agree that the focus on simple properties by much of the existing literature in general is an issue for obtaining models that are directly useful in practice. However in Table 1, each of the 3 datasets have one experimental property that is non-trivial: 1) HIV: HIV virus replication inhibition, 2) BBBP: blood-brain barrier permeability, and 3) BACE: human β-secretase 1 inhibition. For other datasets, we follow standard protocol with a standard set of properties. From a machine learning perspective, the problems tackled by the literature are not solved considering the massive gap in performance between all methods compared to STGG+ and Graph DiT (for example, see Table 2 and 3).
>
> Again, we fully agree that the focus on basic properties is not useful from a chemists perspective, but this is a limitation of the typical datasets and benchmarks used by much of the machine learning community in general. In this work, we chose these datasets because they allow for better comparability to existing work.
>
> > MinMAE is a weak metric - MeanMAE reflects the true capability of a model - Sure, the STGG+ decreased the error to a few magnitudes lower, but this seems more of a float-point accuracy improvement
>
> - Table 2 experiments with the 3 datasets evaluate the Property Accuracy on HIV, BBBP, BACE (the meaningful properties mentioned above) and the MAE (i.e., MeanMAE) on the synthetic accessibility (SAS) property. We show strong performance improvements over recent state-of-the-art methods (far beyond float-point accuracies). As mentioned, we do not use the MeanMAE in Table 3 because we cannot replicate the baselines by Kwon et al. (2023). The authors of this paper themselves told us that they don’t have access to the code anymore, thus replication is not possible so we used their previous metrics which are Efficiency and MinMAE.
> - Following your suggestion, in Appendix A.12, we added the average MAE over the top100 molecules for Table 3 and 5. Table 5 use 100 molecules, so this is the MeanMAE. Since Kwon et al. (2023) did not release code, we can only report on STGG and STGG+ models.
>
> > the authors picked some values that can be exactly achieved by certain molecules and STGG+ found one.
>
> We want to clarify that we used +-4 standard-deviation, as mentioned in line 433. These are the exact same values as used by Kwon et al. (2023), we did not pick the values manually.
>
> > Sure, the STGG+ decreased the error to a few magnitudes lower, but this seems more of a float-point accuracy improvement
>
> - Since a few reviewers were wondering about the difficulty of the OOD task, we added a row in both Tables 3 and 5 with the training data closest sample where we see that except for high QED, all +-4 SD OOD properties are far away from the closest sample in the training data: our model generate molecules much closer to the desired OOD property than the closest training data molecules.
>
> Table 3 (STGG+ is much better except for QED-high where we perform slightly worse):
> | | molWt-low | molWt-high | logP-low | logP-high | QED-low | QED-high |
> | --- | --- | --- | --- | ---  | --- | --- |
> | MinMAE from closest training data samples | 5.7e+1 | 7.3e+1 | 1.5e−1 | 2.0e−3 | 1.8e−2 | 8.2e−4 |
> | MinMAE from the best STGG+ samples | 1.0e−3 | 6.1e−3 | 2.0e−7 | 1.6e−3 | 7.0e−6 | 1.2e−3 |
>
> From 5.7e+1 to 1.0e−3 and 7.3e+1 to 6.1e−3 are more than floating point accuracy improvements.
>
> Table 5 (STGG+ is much better except for QED-high where we perform equally):
> | | molWt-high | logP-low | logP-high | QED-high |
> | --- | --- | --- | --- | ---  |
> | MinMAE from closest training data samples | 1.40 | 9.62 | 0.17 | 0.01 |
> | MinMAE from the best STGG+ samples | 0.47 | 0.35 | 0.01 | 0.01 |
>
> From 1.40 to 0.47 and 9.62 to 0.35 are more than floating point accuracy improvements.
>
> [1] Jang, Yunhui, Seul Lee, and Sungsoo Ahn. "A Simple and Scalable Representation for Graph Generation." The Twelfth International Conference on Learning Representations. 2023.
>
> [2] Ahn, Sungsoo, et al. "Spanning tree-based graph generation for molecules." International Conference on Learning Representations. 2021.

---

> > ### Comment · Reviewer_WwA3 · 2024-12-03
> >
> > Thanks for the additional results.
> >
> > I still do not like the idea of "masking invalid tokens" - the overhead does not seem to be worth it. Other models such as Graph GA  shown in Table 2 and evolutionary algorithms can already generate valid molecules very well. In addition, if you are checking valencies along the way, you can also check MolWt, logP, etc., along the way with a similar overhead of checking valencies. The transformer should learn such grammar by itself for efficiency.
> >
> > The HIV/BBBP/... metrics are indeed more interesting than MolWt, but still, they are evaluated with a random forest regressor. In other words, these properties are still not challenging enough to learn unless their values are based on experimental results. Plus, baseline "Graph DiT" also showed great performance. The quantum chemical properties in the QM9 dataset [1] are popular baselines in many works - HOMO-LUMO gap is one of them.
> >
> > The current results did not include evolutionary algorithms as baselines. I think these methods are more fit and efficient for the metrics discussed in this work. Take this 1995 publication as an example: [2]. While this method might be a bit old-schooled and not so much in the current trends, it is great for OOD generation.
> >
> > For the MinMAE, the authors still did not show the mean MAE over the 1000 generated molecules. Even though I acknowledge that it might not be feasible to report such metrics on the baseline methods, I am still interested in finding out the performance of STGG+ itself. Picking the top 100 is not convincing. In real-life applications, generating 1000 molecules and labeling them can be unfeasible.
> >
> > In summary, I do not believe that the community would benefit much from the STGG+ improvements compared to SMILES. The idea of "any-property conditional" generation is what I find most valuable in this work, but the presented properties and results are not convincing enough for me. This work also lacks chemistry expertise - for example, the generated molecules in figures 4, 5, 6, and 8 do not seem quite valid to me. They are not charge-balanced and some have invalid geometries (e.g., Figure 8). Due to my existing concerns, I will keep my original score.
> >
> > [1] Ramakrishnan, Raghunathan, et al. "Quantum chemistry structures and properties of 134 kilo molecules." Scientific data 1.1 (2014): 1-7.
> > [2] Venkatasubramanian, Venkat, King Chan, and James M. Caruthers. "Evolutionary design of molecules with desired properties using the genetic algorithm." Journal of Chemical Information and Computer Sciences 35.2 (1995): 188-195.

---

### Official Review · Reviewer_mZLQ · 2024-11-04

**Soundness:** 2
**Presentation:** 3
**Contribution:** 2
**Rating:** 5
**Confidence:** 4

**Summary:**

This paper proposed to extend Spanning Tree-based Graph Generation (STGG) to multi-property conditional generation, namely STGG+, with improvements to successfully engineer and implement the concept. STGG+ achieves SOTA on both in-distribution and OOD conditional generation, as well as reward maximization.

**Strengths:**

- This paper is presented with sufficiently clear descriptions.
- The authors explored a wide range of techniques that can be applied in the under-explored context of multi-property conditional generation.

**Weaknesses:**

- It seems to me that the authors invented complicated ad-hoc designs and specifically engineered to fix any issues that may arise, for example by masking the creation of rings when reaching max number (100), or alternating the use of CFG and ranking via a property-predictor. I'm afraid this hampers the overall generality of the proposed method.
- Ablation studies are missing. What's the effect of the improved Transformer architecture against the vanilla one? How does the auxiliary property prediction loss contribute to the results? The same applies to CFG w/ and w/o random guidance, the masking mechanism, the ring overflow treatment, the order randomization, and the automatic construction of vocabulary instead of a predefined one. Detailed ablations are needed to validate the authors' special designs, and provide more insight to the community.

**Questions:**

In Table 2, why not report the MAE at different percentiles, instead of only the MinMAE? It's possible that the model simply memorizes some extreme cases seen in training so as to achieve a good minimum MAE.

---

> ### Author Response · Authors · 2024-11-15
> **mZLQ**
>
> Thank you for your review. We address your questions and comments below.
>
> Weaknesses:
> 1) These choices are applicable to molecules in general, we generally do not change most hyperparameters or modeling decisions between datasets. Making such choices is common to many generative models across domains, such as the need to set a maximum size or number of tokens. As such, these decisions do not affect the generality of the method to molecules.
> Also note you can ignore the self-criticism through the property predictor by simply setting k=1, which we evaluate throughout the paper.
>
> 2) Please refer to Table 8 of Appendix A.8 for ablation analysis, which already addresses most of your suggestions. We made this more prominent in the updated version.
>
> 3) We agree that it would be preferable to show the MAE with percentiles. However, we reached out to the author of the VAE baselines and they didn’t have access to their code anymore. Replicating is non-trivial, so for comparison, we focused on the MinMAE which is the only metric used in the paper with the VAE baselines. In most cases, it's impossible for the model to memorize extreme molecules, see the % of molecules below and above mean + 4SD below:
>
> For Zinc:
>
> MolWt, number of observations above u+4sd = 0
>
> MolWt, number of observations below u-4sd = 0
>
> MolLogP, number of observations above u+4sd = 0.0004% (1 case)
>
> MolLogP, number of observations below u-4sd = 0.068%
>
> QED, number of observations above u+4sd = 0
>
> QED, number of observations below u-4sd = 0.019%
>
> For Chromophore:
>
> ExactMolWt, number of observations above u+4sd = 0.67%
>
> ExactMolWt, number of observations below u-4sd = 0
>
> MolLogP, number of observations above u+4sd = 0.78%
>
> MolLogP, number of observations below u-4sd = 0
>
> QED, number of observations above u+4sd = 0
>
> QED, number of observations below u-4sd = 0

---

> > ### Author Response · Authors · 2024-11-27
> > **Response 2**
> >
> > Since our previous response, we added content to the paper that should further address some of your concerns. Please let us know if your concerns are now addressed.
> >
> > > MAE at different percentiles
> >
> > In Appendix A.12, we added the average MAE over the top100 molecules for Table 3 and 5 in addition to the existing MinMAE. As mentioned in the previous message, Kwon et al. (2023) did not release code, thus we can only report on STGG and STGG+ models.
> >
> > > It's possible that the model simply memorizes some extreme cases seen in training so as to achieve a good minimum MAE.
> >
> > Since a few reviewers were wondering about the difficulty of the OOD task, we added a row in both Tables 3 and 5 with the training data closest sample where we see that except for high QED, all +-4 SD OOD properties are far away from the closest sample in the training data: our model generate molecules much closer to the desired OOD property than the closest training data molecules.
> >
> > Table 3 (STGG+ is much better except for QED-high where we perform slightly worse):
> > | | molWt-low | molWt-high | logP-low | logP-high | QED-low | QED-high |
> > | --- | --- | --- | --- | ---  | --- | --- |
> > | MinMAE from closest training data samples | 5.7e+1 | 7.3e+1 | 1.5e−1 | 2.0e−3 | 1.8e−2 | 8.2e−4 |
> > | MinMAE from the best STGG+ samples | 1.0e−3 | 6.1e−3 | 2.0e−7 | 1.6e−3 | 7.0e−6 | 1.2e−3 |
> >
> > Table 5 (STGG+ is much better except for QED-high where we perform equally):
> > | | molWt-high | logP-low | logP-high | QED-high |
> > | --- | --- | --- | --- | ---  |
> > | MinMAE from closest training data samples | 1.40 | 9.62 | 0.17 | 0.01 |
> > | MinMAE from the best STGG+ samples | 0.47 | 0.35 | 0.01 | 0.01 |

---

### Official Review · Reviewer_GMmJ · 2024-11-04

**Soundness:** 3
**Presentation:** 2
**Contribution:** 2
**Rating:** 6
**Confidence:** 3

**Summary:**

The authors build upon spanning tree-based graph generation methods to produce valid molecules with desired properties. They enhance the original network architecture by adding property embeddings and incorporate a properties predictor head for joint training. Through the use of classifier-free guidance and conditioning on these properties, the authors demonstrate that STGG+ can generate molecules conditioned on specific properties or with high reward values.

**Strengths:**

1. To my knowledge, this manuscript is the first to thoroughly examine STGG for reward conditioning and/or optimization.
2. The work reflects a substantial effort to assess STGG+'s capabilities. Overall, the approach appears methodologically sound.
3. Molecular property optimization is an open challenge. Given its competitive performance compared to existing algorithms and its use of a (somewhat) unique molecular representation, I expect this work will attract reasonable interest.

**Weaknesses:**

1. The primary limitation of this paper is that generating 'valid' molecules does not guarantee synthesizability. Many molecules presented in the appendix would be very challenging, if not impossible, to synthesize. Meanwhile, some baseline methods may perform slightly worse on reward but produce molecules that are easier to synthesize, avoiding "reward hacking." A fairer comparison would involve evaluating the reward optimization performance of synthesizable molecules across different algorithms.
2. While novelty is not an ideal measure of a paper's value, this work is highly empirical, with limited theoretical insight. This puts more emphasis on competitive performance, yet the paper lacks adequate baseline comparisons for reward optimization. Previous work has shown that methods such as GraphGA, LSTM-HC, and Reinvent are effective at maximizing OOD reward, and these baselines should be included in Sections 4.3–4.5 (particularly Section 4.5, which currently lacks any baseline comparison). This is especially relevant as the random guidance approach for OOD generation resembles slightly enhanced random sampling with a reward proxy.
3. The molecular properties selected for optimization in this study are _very_ simple. For instance, molecular weight can be adjusted by adding or removing atoms, and logP by incorporating ionic groups (which the model does). Optimizing HOMO-LUMO gaps within the QM9 dataset is not useful, as these molecules contain only 9 atoms. These problems are generally considered solved.
4. Although the work is extensive, certain details are presented inconsistently or lack substantiation. Some claims are unsupported by data (e.g., statements like "other [configurations] were not beneficial/did not improve performance" lack any data references). Additional issues include appendix figure captions that are unclear and lack cross-references in the main text (e.g., molecules with QED > 1 in figures 8 and 14), and captions inappropriately implying low QED correlates with implausibility (e.g., figure 9). Many terms are undefined, including "synthe. MAE," "HOMO/LUMO," "SAS," as well as the precise definition of diversity used here. Additionally, error bars are missing in all tables.

**Questions:**

1. In SMILES notation, the same molecule can have multiple string representations. Based on my (limited) understanding of STGG, this ambiguity seems present as well. How are the molecules canonicalized?
2. In Tables 3 and A.10, do all methods reach peak performance only after generating 1 million molecules? Is the search space the same across methods?

---

> ### Author Response · Authors · 2024-11-15
> **GMmJ**
>
> Thank you for your review. We address your questions and comments below.
>
> Weaknesses:
> 1) The main goal of our work is to show that our model is better able to generate molecules of the specified properties to condition on. One such property could be synthesizability. We agree that synthesizability is an important problem in molecule generation, but good metrics of synthesizability are difficult to obtain. Therefore, we focus on improving the property conditioning of properties in general, and with a suitable dataset that contains synthesizability as one of the properties, we should be able to improve generation of synthesizable molecules as well.
>
> We will add comparisons of synthesizability (determined by AiZynthFinder) between training molecules (since synthesizability depends on the specific choice of reactants and starting molecules chosen, it's likely that many training molecules are labeled as not synthesizable, especially for ChromophoreDB), generated molecules in-distribution and out-of-distribution in time for the camera-ready-version.
>
> 2) We accounted for GraphGA, LSTM-HC in Table 1 (we had to trim the table as it was huge, the full table with many more baseline methods is in Appendix A.9) and they are not particularly good on the 3 datasets of that table. We used numbers from “Liu, Gang, et al. Graph Diffusion Transformers for Multi-Conditional Molecular Generation. arXiv preprint arXiv:2401.13858 (2024).”, so implementing those baselines into our code for our other tasks would take significantly more work. We believe that the much worse performance on 3 datasets is enough of a filter to justify not pursuing these methods further in the other datasets.
>
> 3) We fully agree that the focus on simple properties by much of the existing literature in general is an issue for obtaining models that are directly useful in practice. However in Table 1, each of the 3 datasets have one experimental property that is non-trivial:  1) HIV: HIV virus replication inhibition, 2) BBBP: blood-brain barrier permeability, and 3) BACE: human β-secretase 1 inhibition (mentioned in line 361-363).
> For other datasets, we follow standard protocol with a standard set of properties. From a machine learning perspective, the problems tackled by the literature are not solved considering the massive gap in performance between all methods compared to STGG+ and Graph DiT (for example, see Table 1 and 3).
>
> 4) Thank you for catching these issues. We added more details, cross-references, and fixed incorrect statements to the items that you mentioned.
>
> Questions:
>
> 1) Contrary to STGG, we do not canonicalize molecules, we use a random ordering of the molecules (a different random ordering is sampled for each molecule during training).. Doing so improves generalization (see Table 8 in Appendix A.8). It was only mentioned on line 85 in ‘contributions’ that we used random ordering; we added a mention in the text because it should have been made more clear.
>
> 2) The non-STGG results are from “Jain, Moksh, et al. "Multi-objective gflownets." International conference on machine learning. PMLR, 2023”; they use 1M molecules, they do not show performance over time. The search space is parametrized differently, but all methods have the same output space.

---

> > ### Comment · Reviewer_GMmJ · 2024-11-24
> > **Official Comment**
> >
> > I thank the authors for their responses to my concerns. Some of my questions are resolved. Nevertheless, I am a bit puzzled by why random guidance (random sampling with a reward proxy?) performs better than GraphGA or other baselines; it'd be great if the authors can elaborate on it.
> >
> > I am willing to raise my score to 6 with the revisions implemented as I see merit in this work.

---

> > > ### Author Response · Authors · 2024-11-25
> > > **Response**
> > >
> > > To clarify, the experiments on Table 2 (which compares to GraphGA and other baselines on 3 datasets) do not use random guidance.
> > >
> > > Random guidance is only used for out-of-distribution (OOD) generation (Table 3 and 5). The rationale is that strong guidance can become problematic when conditioning on extreme OOD values (which we sometimes observed, see lines 446-448), and random guidance allows the model to still perform well when high guidance becomes problematic for certain properties (see lines 279-286) because the model generates at various levels of guidance. Furthermore, in combination with the self-property-predictor (k=5), it means that we try different guidance levels and the property-predictor can automatically infer the best guidance level since it returns only the molecules with the best estimated properties (lines 284-286).
> > >
> > > Random guidance can be seen as a way to vary exploration and exploitation (higher guidance means less diversity and better property-alignment, while lower guidance means more diversity and less property-alignment). We now added this important information in lines 286-288.

---

> > > > ### Comment · Reviewer_GMmJ · 2024-11-25
> > > > **Comments**
> > > >
> > > > I have updated my score. I would appreciate the authors could highlight the revision changes in the main paper and appendix.

---

> > > > > ### Author Response · Authors · 2024-11-26
> > > > > **highlights**
> > > > >
> > > > > Thank you. We updated the main document to contain all the highlights, including those in the appendix.

---

### Official Review · Reviewer_eLzS · 2024-11-10

**Soundness:** 3
**Presentation:** 3
**Contribution:** 3
**Rating:** 6
**Confidence:** 2

**Summary:**

The paper presents STGG+, a model for molecule generation with conditional property control. The model also has the ability of self-criticism to select optimal outputs. It achieves high validity and diversity in generated molecules, efficiently handling both typical and extreme properties.

**Strengths:**

The paper presents molecule generation models, allowing multi-property control and self-assessment of generated molecules. It’s well-designed, with detailed experiments showing strong results across different datasets. The writing is clear and structured.

**Weaknesses:**

The self-criticism mechanism for filtering generated molecules based on property predictions is a key feature, but there is limited evaluation of its accuracy. Detailed analysis will be necessary.

I am not an expert of this field so I will lower my confidence score.

**Questions:**

In the OOD setting, how the model’s performance varies under different guidance settings, especially for extreme or non-physical property values?

---

> ### Author Response · Authors · 2024-11-15
> **eLzS**
>
> Thank you for your review. We address your questions and comments below.
>
> To clarify, we already provide property prediction performance of the self-property-predictor in Table 6 of Appendix A.6. For generation, we always compare with and without self-criticism in all experiment tables (k=1 vs k>1).
>
> To answer your question,we found that the self-property-predictor was not always optimal at OOD and can make mistakes when choosing the best-out-of-k=5; we found this to happen for high logP conditioning values (we mention this in lines 440-443 and 534-535).

---

> > ### Comment · Reviewer_eLzS · 2024-11-26
> >
> > Thank you for the response, and my concerns have been addressed. I keep my rating and confidence unchanged.

---

### Author Response · Authors · 2024-11-27
**Improvements to the paper**

Having seven reviewers, it was quite challenging to navigate the many reviews. But, we worked hard on addressing the concerns of all reviewers.
Since the discussion, we have improved the paper as follows:
- changed Fig 1 to highlight our contributions over STGG
- mention that we randomize the order of the molecules and that this increases generalization over STGG
- explain better the random guidance through balancing exploration and exploitation
- simplified Figure 2 for better understanding
- clarified which property predictor is used for each property (RDKIT unless otherwise specified)
- added the unconditional generation table (Table 1) to the main paper
- cleaned Table 2 headings and details to make clear what are the metrics and how they work
- added a row to Table 3 and 5 showing the "Training data (closest real sample)" for the MinMAE. This shows that no training sample is close to the +-4SD desired properties, thus our model extrapolates well beyond what is known.
- added information in the Appendix A.3 on canonicalization (how STGG molecules are converted back to canonical SMILES)
- added details on how the property masking works in the Appendix A.3
- discuss other potential architectures that we also considered in the Appendix A.5
- added the top-100 MAE results for Table 3 and 5 in Appendix A.12
- revised a few typos

We believe that most concerns have been addressed and that the paper has been significantly improved through this discussion.

Please consider revising your score if you feel that your concerns have been well addressed.

---

### Author Response · Authors · 2024-12-04

Dear Reviewers and Area Chair,

As the discussion period concludes, we want to thank all reviewers for their constructive feedback. We have worked hard to address all concerns, which has significantly improved our paper.

The reviews recognized our effort to present a well-structured paper with sound methodology and novel contributions in an under-explored topic. At the same time, some criticisms—e.g.,  questions regarding the generalizability of our method to more complex properties or the chemical validity of the generated molecules—struck us as quite severe. While we acknowledge the intrinsic limitations of addressing such a complex and evolving problem, we feel that certain expectations of impact and generalizability exceed what is realistically feasible in this domain at this stage.

By advancing the state of the art and providing rigorous assessments, we believe our work is both valuable to the community and suitable for publication at ICLR.

We thank you again for your engagement and feedback throughout this process.

Best
The authors

---

### Meta-Review · Area_Chair_4xrT · 2024-12-22

**Metareview:**

**Summary**

This work extends an unconditional molecular generation method ‘Spanning Tree-based Graph Generation’ (STGG) to enable conditional generation given some desired properties. Toward that end, the authors enhance the Transformer architecture of STGG to include (a) random masking of subsets of properties during training, (b) classifier-free guidance to generate multiple conditional samples, (c) a property-predictor component, and (d) a self-criticism mechanism to filter out from the generated molecules the ones whose predicted properties are not aligned with the desired values.  The proposed method, called STGG+, is empirically shown to achieve strong performance on reward maximization besides in-distribution and out-of-distribution (OOD) conditional generation.


**Strengths:**

Reviewers appreciated different aspects of this work such as (a) practical relevance of the setting for molecular generation, reward conditioning, and optimization, (b) methodological soundness, (c) clear writing and presentation, (d) detailed experimental results, and (e) self-criticism as a nice capability,.

**Weaknesses:**
Many concerns were raised by the reviewers. These included (a) validity of a generated molecule being insufficient to guarantee its synthesizability (hence the need to benchmark the methods on the performance of their respective synthesizable molecules), (b)  limited theoretical analysis or insight, (c) missing stronger baselines for OOD reward, (d) generated molecules being too simple, (e) unsubstantiated empirical claims, (f) possible over-engineering, (g) missing confidence intervals or error-bars, and (g) lack of ablation studies.  Questions were also raised about choice of some metrics and hyperparameters, conveying concerns about the generality of the proposed method as well its potential vulnerability to memorise extreme cases from the time of training.


**Recommendation:**
During the discussion period, some concerns were addressed by the authors. However, some reviewers maintained that some major issues remained unresolved.

In particular, reviewer GMmJ asserted that the concerns about synthesizability were significant since many of the molecules generated with STGG+ seemed hard to synthesize. They also pointed to possible misalignment between reward and the ability of the different methods to generate easy-to-synthesize molecules. They also questioned whether the benefits of STGG+ were statistically significant absent the error bars.  I fully agree with these concerns.

Similarly, Reviewer WwA3 made several critical observations. They pointed the flaws inherent in masking invalid tokens: (a) marginal performance gains did not justify the enormous overhead, and (b) STGG+ should have been able to efficiently learn to generate valid molecules intrinsically. They also pointed out to insufficiency of some experiments and reporting of experimental results: e.g., they pointed out (a) mean MAE was not shown for a statistically significant number of generated molecules), (b) values of properties such as HIV/BBBP were not based on experimental results but predicted using a random forest regressor, and (c) comparisons were not provided against  relevant evolutionary algorithms, and d) STGG+ seemed  to produce invalid generated molecules (showing charge imbalance and invalid geometries). Reviewer atbB also agreed with this assessment.  I echo these critical concerns.

Based on the concerns raised by the reviewers, I decided to closely review the manuscript myself as well, and discovered further critical issues with the current submission.  The current state of the art in this area comprises methods such as JODO, EEGSDE, TEDMol, EDM, Modular Flows, GeoLDM, GeoBFN, and Twigs. All these methods can already achieve “any-property-conditioning”, so I’m afraid this cannot be claimed as a novelty at all. Most of these methods - with the exception of Twigs - have been around for a while, still they have not been mentioned in related work or compared against empirically.

In fact, several of these methods are capable of generating 3D molecular conformers, beyond the capabilities of STGG+ (which is restricted to 2D generation - and this also explains in part several unresolved issues with STGG_ that many reviewers pointed out). Unless convincing experimental benefits are demonstrated against many of these strong baselines (prior published work seems to indicate that most of these baselines outperform STGG+ across metrics),  it’s hard to vote for this paper. Therefore, I recommend clear rejection.

**Additional Comments On Reviewer Discussion:**

I commend the reviewers for their exceptional service - most of them provided detailed and thoughtful feedback, and engaged in the discussion. All the key points of discussion, and how they helped inform the recommendation, have already been included above.

---

### Decision · Program_Chairs · 2025-01-22

Reject